# Cover versions as an impact indicator in popular music: A quantitative network analysis

José Luis Ortega [1,2] *

1 Institute for Advanced Social Studies (IESA-CSIC), Córdoba, Spain, 2 Joint Research Unit Knowledge Transfer and Innovation, (UCO-CSIC), Córdoba, Spain

* jortega@iesa.csic.es

## Abstract

In contemporary popular music, covering or adapting previous songs signals a tribute or reworking of popular hits, which implies acknowledgement of the original musicians. This connection can be interpreted as a form of musical impact among artists. A network graph with more than 106k artists and 855k cover versions extracted from the web site Second-HandSongs was created. The objective is to explore the shape of this network, identify the most relevant artists according to different impact measurements and to visualize connections between music genres. This analysis is done from a longitudinal perspective with the aim of understanding how cover versions can inform us about the history of the contemporary popular music. Results show that the number of covers by artist is skewed distributed, diminishing gradually since the 1950s. Different network metrics have allowed to identify the most covered (weighted indegree), the most influential (PageRank) and the most crossover artists (weighted betweenness centrality). The network graph also shows that genre affinity is the main criterion for covering songs between artists, language being the second. Remakes from other genres reflect that *Jazz* and *Pop/Rock* are the most influential because they emerge stronger and form the core of their respective sub-networks. Cover songs describe two cycles. In a first phase from 1900s to 1950s, dominated by Jazz and Vocal artists, the covers are more frequent and associated with the notion of reworking (e.g. jazz standards); in a second stage, since the 1950s, when the Pop/Rock emerges, cover songs are less common and seen as tribute.

## Introduction

Popular music has become one of the most important cultural phenomena in the 20th and 21st centuries [1]. The invention of broadcasting media, new recording systems and the establishment of the middle class have favoured the mass consumption of recorded music, making it one of the most fruitful entertainment businesses today [2, 3]. Since the early 20th century, artists and bands have emerged over the years writing and performing thousands of popular songs that reflect the worries, hopes and feelings of society. Local traditions (*Flamenco*, *Samba*,

**Data Availability Statement:** All Cover songs files are available from the Open Science Foundation database (accession URL https://osf.io/6j2ru/).

**Funding:** The author(s) received no specific funding for this work.

**Competing interests:** No authors have competing interests.

*Celtic music*, etc.) and new technological advances (amplifiers, synthesisers, etc.) have led to the proliferation of genres and styles that have enriched musical expression and enabled the transmission of new musical forms to the next generation [4].

One element that reveals the hereditary nature of popular music is the cover, adaptation or remake of songs. A cover is a new recording of a piece of music originally written or performed by another musician. These acknowledgements lead to links between artists and illustrate the musical influences of soloists or bands, while assessing the impact of artists' careers in specific musical communities.

The advent of the World Wide Web ushered in the appearance of specialized music databases that began collecting and publishing information about artists, discs, songs and styles [5]. The main purpose of these services was to promote disc catalogues and to preserve intellectual rights [6]. Allmusic, Discogs and MusicBrainz are just a handful of open music resources that provide detailed information about commercial music recordings. Second-HandSongs (secondhandsongs.com), with close to one million covers, is the most complete database collecting information about cover songs and their performers. All these platforms provide a great volume of data that can foster our knowledge about the contemporary popular music from a quantitative perspective.

This study therefore attempts to address this research gap analysing how cover songs between artists allows us to know the evolution of the popular music over a century, identifying their most influential artists and describing the emergence of genres and styles.

## Related research

Until recently, music studies have been distinguished by the use of qualitative methods such as case studies and textual criticism. The appearance of many web-based music databases and advances in music data processing and analysis have led to the emergence of a new quantitative paradigm in this discipline [7, 8]. From this perspective, Rose et al. [9] used digitized documents to study the rise and fall of music printing in the 16th and 17th centuries, tracking specific composers and observing changing trends in genres. Park et al. [10] studied the co-authorship network of western classical music composers, finding preferential attachment as a growth mechanism. And McAndrew and Everett [11] analysed encounters between British classical composers to detect hotspots of creativity.

Within the scope of contemporary popular music, many papers have studied collaboration patterns between Jazz musicians [12], music industry agents [13] and Finnish heavy metal artists [14]. Other studies have focused on lyrics content. Napier and Shamir [15] examined the sentiment of 6.1k songs from 1951 to 2016, finding that anger and fear have increased significantly. Ruth [16] observed that love is the most relevant topic in German rock music. However, the most common research theme is audio features analysis to observe changes and tendencies. Collins [17] studied the influence of early electronic dance music through audio content analysis. Mauch et al. [18] analysed the harmonic and timbral properties of 17k Pop and Rock songs to detect the evolution of styles. Klimek et al. [19] compared the musical styles of eight million music albums to test whether new symbols are introduced by current elite members. Finally, Youngblood [20] was based on Whosampled site to test the conformity bias in the cultural transmission of music samples.

Nevertheless, an important gap in the quantitative studies on the contemporary popular music has been the total absence of analysis on cover versions. In most of the cases, they have been observed from a theoretical perspective, figuring out the cultural meaning of this type of songs. For Plasketes [21], covers are a symptom of standardization and loss of originality, while others [22, 23] defend the importance of covers in the transmission of styles and ideas.

In this context of cultural studies, many papers have explored the meaning of cover versions using specific cases studies. Doktor [24] analysed cross-gender adaptations of the song *Satisfaction*. Chu and Leung [25] studied the role of cover songs in the decline of Cantopop. The only exception is the work of Anderson [26] who quantified the 'adaptations' wave in French Pop.

## Objectives

The principal objective of this study is to visualize and analyse the cover songs network between artists in order to describe its topology and evolution throughout the time. In this form, it attempts to understand how the use of cover versions has changed over the decades, what artists have been more influential in each moment and how the different music trends have been associated between them. To that end, several network indicators (weighted indegree, PageRank and weighted betweenness centrality) are proposed to assess distinct forms of impact on the artists. Three research questions were formulated:

- What are the topological characteristics of the cover songs network and how do they evolve over time?

- Which musical artists are the most influential in the network and how have they reached that position? What information does each network metric provide about the impact?

- Can cover songs inform us about the context of each music genre? Is it possible to observe the evolution of the preferences for each genre?

## Methods

### Data sources

**SecondHandSongs.**   This database records data about covered songs and their performers and was created by Bastien De Zutter, Mathieu De Zutter and Denis Monsieur in 2003. Since then, the database has been supplied by a group of volunteer editors that manually add and update the information about songs and artists. As a result, the database coverage may not be exhaustive or uniform. In January 2020, the site reported 788k covers, 143k artists and 96k original songs. According to SecondHandSongs, original performers are those that perform, record or release a song for the first time, and a cover is when subsequent artists perform, record or release that same song again. This definition excludes songwriters, who could be the original authors, but not the original performers. This database is limited to covered songs only and does not provide a comprehensive list of recorded songs by artists [27].

**Allmusic.**   An exhaustive music database released on the Web in 1994 and owned by TiVo Corporation. At the last estimate it provided 30 million tracks and three million albums [28]. Allmusic's most valuable information is the classification scheme. Each artist is arranged according to 21 genres and over 100 styles. Styles are not considered as sub-categories of genres, therefore, they are labels or keywords freely assigned to an artist.

### Data extraction

Web scraping techniques were used to obtain data from SecondHandSongs and Allmusic. WebQL software was used to write ad-hoc scripts that crawl and retrieve specific information from each site. Details about songs, covers and artists were retrieved from SecondHandSongs. Each performer is identified by a sequential number. Then, numbers from 1 to 175000 were automatically generated to retrieve the list of covered songs and artists by each

performer. (This process was carried out in January 2020 and the data are accessible in: https://osf.io/6j2ru/).

Because SecondHandSongs only includes information about covered songs, Allmusic was used to broaden information about artists. Concretely, the complete number of songs recorded by each artist and the genres and styles of each performer. The total number of songs allows us to measure an artist's production, while the genres and styles enable us to group artists. As with SecondHandSongs, sequential codes from mn0000000001 to mn0003600000 were queried to retrieve artist profiles in Allmusic, obtaining 176k.

## Data cleaning

SecondHandSongs assigns an id to each cover, but not to a song because the title of the cover could be different from the original (i.e. translations, reworking of lyrics). Links between artists were therefore established according to the number of times an artist covers another, regardless of the title of the song. In the event that an artist covers the same song more than once, it was counted as a different cover.

The name of the artist or band was used to match SecondHandSongs and Allmusic records. This element was carefully cleaned to avoid mismatches between slight name variations. Thus, some characters such as ampersands and punctuation marks were removed or replaced by normal letters (Simon & Garfunkel by Simon and Garfunkel, "Big Mama" Thornton by Big Mama Thornton). In other cases, names with different variations were normalized (Booker T. and the Memphis Group's by Booker T. and the M.G,'s, Johnny Burnette Trio by Johnny Burnette Rock and Roll Trio); synonymy between different artists (Tom Jones, 60s vocal artist and Tom Jones, 50s lyricist) was solved adding a digit to the name.

Medleys pose another problem because they include parts from more than one song. SecondHandSongs includes this type of song but does not properly identify which performer is the author of each different part. These songs (11,112) were removed because they were not significant (1%) and could introduce a lot of noise.

When a song is traditional or the original performer is unknown, SecondHandSongs assigns it to a generic *unknown* (20,428). These covers were removed because there is not a connection between artists. In addition, SecondHandSongs joins the code of two or more artists when they perform together (e.g. Sinatra and Bono, Elvis Presley and the Jordanaires). These cases were duplicated, assigning the same song to each artist. After all these transformations the final sample included 106k artists and 855k cover songs.

## Social network analysis

Covering a song implies a cultural and meaningful connection between the original artist and subsequent performers. The best way to analyse these connections is through Social Network Analysis, the branch of sociology specializing in the structural analysis of social relationships. Social network metrics are used to identify different type of impact:

- Weighted Indegree centrality ($k^{in}$) measures the weighted number of acs incident to a node [29]. In our case, it is equal to the total number of cover songs of one artist. This indicator measures the accumulated impact of an artist in the music community.

- Weighted Outdegree centrality ($k^{out}$) works in the opposite direction and counts the weighted number of arcs directed out of a node. That is, the total number of covers performed by an artist. This metric allows us to assess artists' ability to praise and acknowledge their influences.

- PageRank (*PR*) is the original algorithm used by Google to rank their search results [30]. It is used to assign a relative weight to each node, according to the value of the linking nodes, which transfer a proportional value to the cited node. It was calculated taking weights into account. The PageRank of an artist is an indicator of the importance of the performers that cover their songs. Thus, an artist whose songs are covered by performers with many versioned songs is more important than artist covered by unknown performers.

- Weighted betweenness centrality (*b*) is an indicator that counts the number of weighted shortest paths that pass through a node [31]. The importance of this metric is that it allows the detection of nodes at an intermediate position between distant nodes and groups. According to song remakes, weighted betweenness centrality scores artists that cover or are covered by other artists belonging to different groups. To a certain extent, it identifies crossover artists that inspire or are inspired by different artists from different styles, countries and epochs.

## Results

### General characteristics of the covers network

A network graph was constructed with 106,095 nodes (artists) and 855,650 arcs (cover versions) extracted from SecondHandSongs. The very low density of the graph (*d*<0.01%) shows few connections between artists, suggesting that covering songs is highly selective. This interpretation could be confirmed by the cumulative distribution of covers performed (outdegree) and received (indegree) by artist. They follow a power-law distribution (Fig 1), with an exponent ($\alpha$) of 2.882 for outdegree and 2.937 for indegree. R package poweRlaw was used to create the cumulative distribution functions (CDF) and the maximum likelihood estimation (MLE) method was used to fit the distributions. This type of distributions suggests a considerable imbalance in the number of covers performed/received by artist. For example, the outdegree

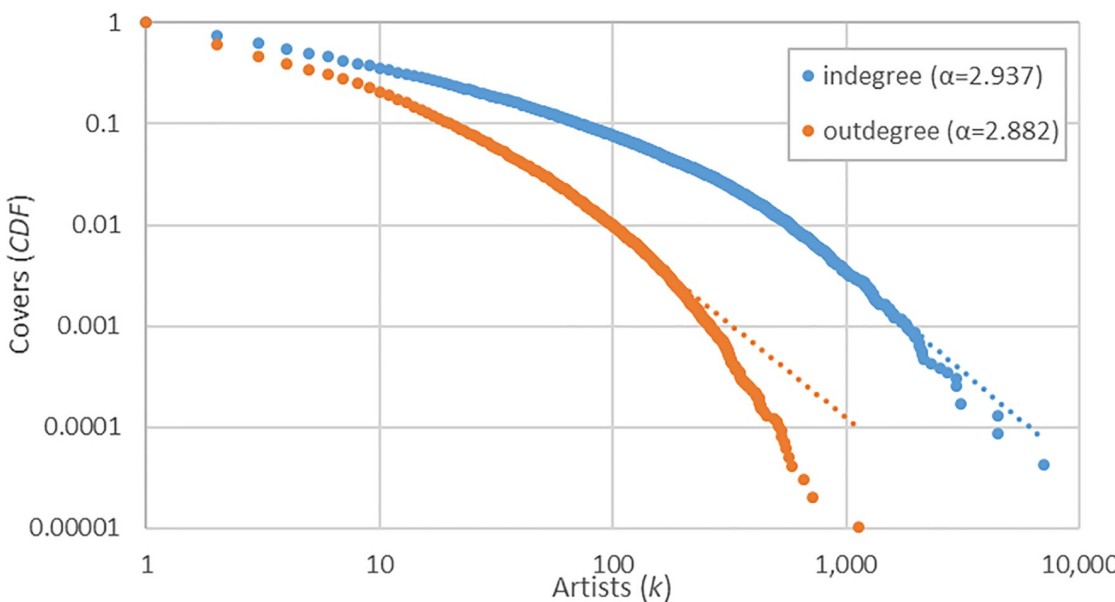

**Fig 1. Cumulative distribution function (CDF) of the number of received (indegree) and performed (outdegree) covers by artist (log-log), and their respective exponents ($\alpha$).**

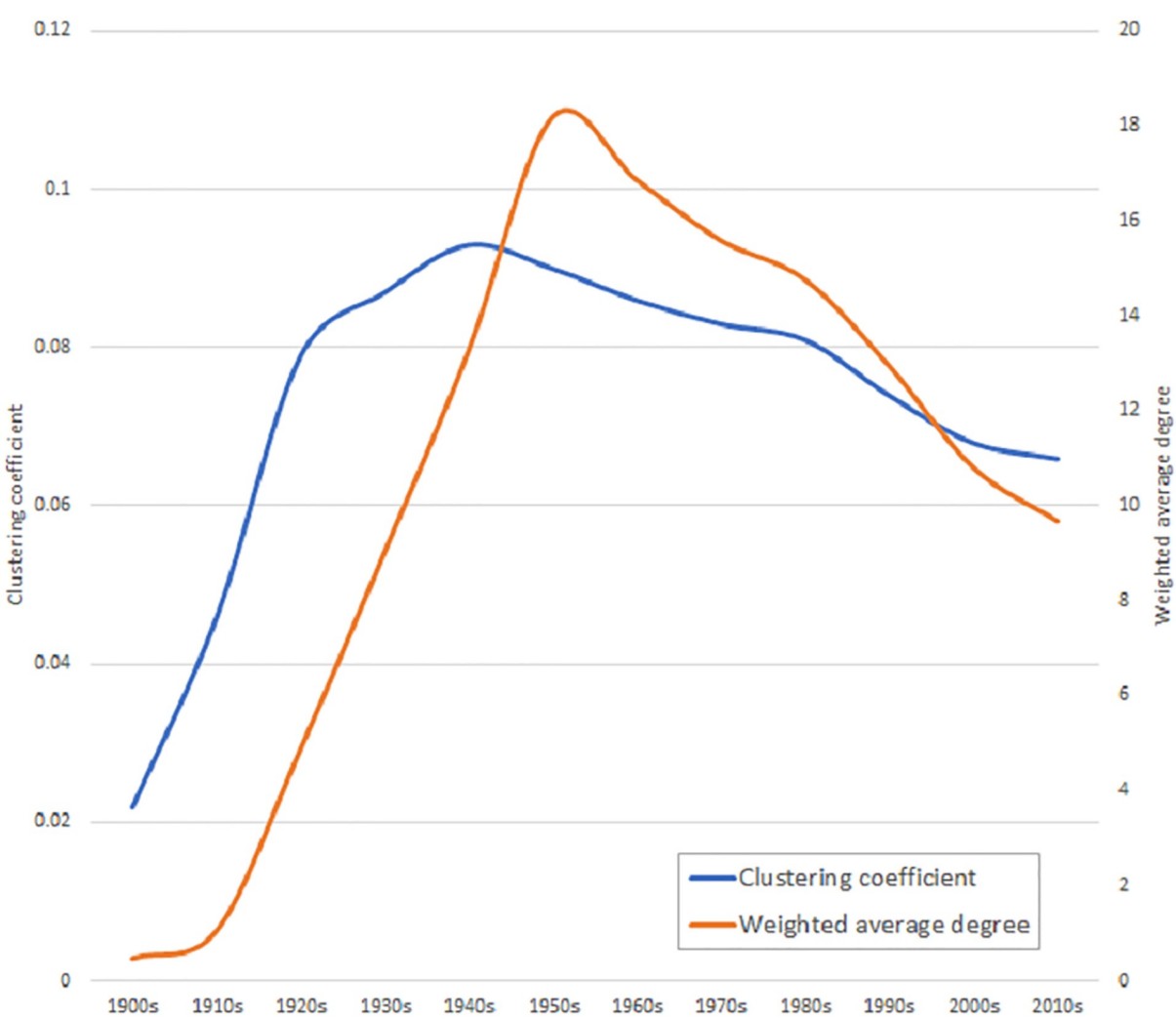

**Fig 2. Evolution of the weighted average degree and clustering coefficient over the decades.**

distribution shows that only 7% of artists perform no cover and that 48% of them cover the songs of at least two artists. These percentages indicate that covering songs is relatively common. However, the indegree distribution reveals that only 22% of the artists have been covered, whereas just 100 artists (0.1%) encompass 41% of all the versions. This high disproportion reveals that while covering songs is a usual way of doing music, these versions are concentrated on songs belonging to a very reduced number of artists who make up the star system of contemporary popular music.

In order to study the evolution of the network, artists were classified by the decade in which they started their musical career. During this process, 4,983 artists were discarded because they did not have that information. Then, several cumulative networks by decade were built. This aggregate approach is because the cover versions are made of current or precedent artists and each network rests on the previous one. The growing pattern of the network is not constant (Fig 2). The average of covers (weighted average degree) strongly increases from 1910s ($k = 1.07$) to 1950s ($k = 18.18$), but since 1960s ($k = 16.85$) this average gradually descends until 2010s ($k = 9.68$) (Table 1). This same occurs with the clustering coefficient, which increases up to .093 in 1940s and then slowly drops to .066 in 2010s. This result suggests that

**Table 1. Network indicators evolution by decade.**

| Network indicators | 1910s | 1920s | 1930s | 1940s | 1950s | 1960s | 1970s | 1980s | 1990s | 2000s | 2010s |
|---|---|---|---|---|---|---|---|---|---|---|---|
| Nodes | 106 | 1353 | 2732 | 4647 | 9735 | 19152 | 27737 | 35502 | 53398 | 80033 | 101112 |
| Arcs | 82 | 5260 | 18974 | 47800 | 138949 | 256180 | 344714 | 419488 | 555932 | 698403 | 792509 |
| Average degree | .774 | 3.888 | 6.945 | 10.286 | 14.273 | 13.376 | 12.428 | 11.816 | 10.411 | 8.726 | 7.838 |
| Weighted average degree | 1.066 | 4.894 | 9.035 | 13.274 | 18.183 | 16.85 | 15.578 | 14.752 | 12.94 | 10.805 | 9.678 |
| Average path length | 3.07 | 3.707 | 3.954 | 3.899 | 3.97 | 4.098 | 4.341 | 4.493 | 4.696 | 4.888 | 4.968 |
| Clustering coefficient | 0.046 | 0.079 | 0.087 | 0.093 | 0.09 | 0.086 | 0.083 | 0.081 | 0.074 | 0.068 | 0.066 |

during the 1910s-1950s, the network is growing in number of covers, with artists that tend to sing previous songs. However, since 1960s, the pattern changes and the artists that joined the network cover less songs.

The covering of songs between generations of musicians can illustrate with more detail this changing pattern. Fig 3 shows how artists that started their career in the 1900s-1950s cover

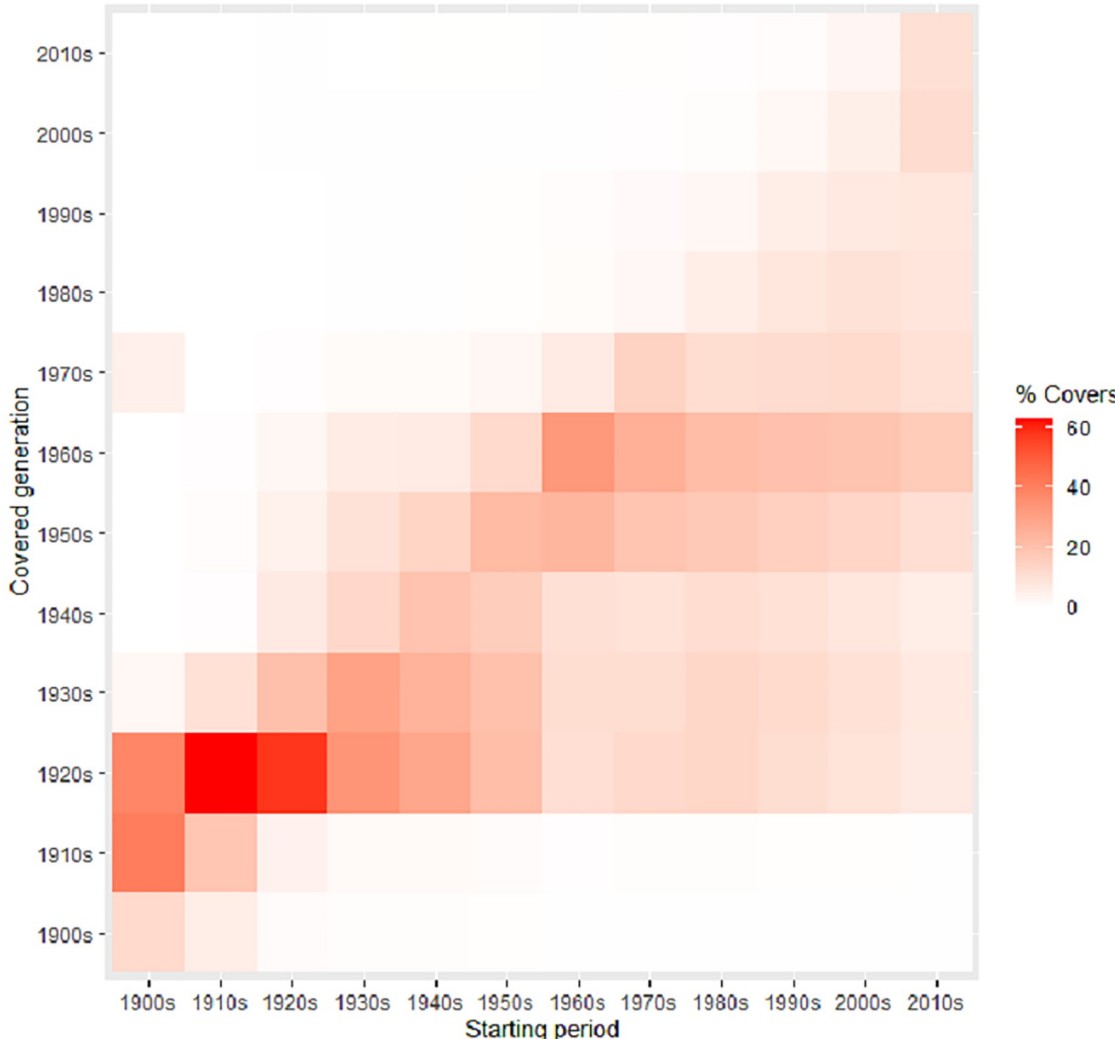

**Fig 3. Heat map with the percentage of covers performed by artists that started their career in certain decade (x) to artists from other decades (y).**

more frequently songs of 1920s artists. This fact would indicate that during the first middle of the 20th Century, artists from the 1920s exercised a great influence on their partners until the 1950s. However, from this decade, it is observed that artists start to cover songs mainly from the 1950s-1960s, suggesting a generational change in the preferences of the artists. This pattern is maintained until the 2000s, when a new change is surmised. One of the possible explanations of these variations would be the emergence of new music genres in specific moments (see below, *Genres and styles* section).

## Top covered artists

The cumulative distribution of covers has showed that just a small portion of artists are profusely covered. This section analyses who are these figures and how they have evolved. Supplementary material S1 File includes the full list of artists and impact indicators. Table 2 presents the ranking of the first ten most versioned artists and bands. Because an artist may cover different songs from the same performer, weighted indegree is used to compute the strength of this association. The purpose is to show who shapes the core of the network and what features they have in common. The number of songs (supplied by Allmusic) contains the total number of musical track recorded and released by artist, including different recordings of the same song. Due to this and the skewed distribution of cover songs (Fig 1), this data was not used as normalization variable. The Beatles is the most covered group with 16,825 versions, followed by the Jazz composer Duke Ellington with 10,564 and the singer Bing Crosby with 7,011. All the performers belong to the English-speaking culture, revealing the strong and early development of the music industry in these countries [32]. Another particularity is that, apart from The Beatles, Bob Dylan and Elvis Presley, they all started their music careers in the 1920s, 1930s and 1940s, which could indicate that there is a time-dependence in the covering of songs. In addition, these old artists belong to genres with a long-standing tradition such as Jazz and Vocal, while The Beatles, Bob Dylan and Elvis Presley come from Pop/Rock.

A more detailed view about the evolution of the impact of these artists is plotting the number of covered songs throughout the decades (Fig 4). Performers from the 1920s–1930s depict a constant growth, headed by Duke Ellington and Bing Crosby, pillars of the Jazz and Vocal music respectively. It is also possible to appreciate the strong emergence of Thelonious Monk and Miles Davis, two jazz figures representative of the Bebop in the 1940s. However, the most significant increase is seen in The Beatles and Bob Dylan, who start their career in the 1960s and in just three decades they climb to the first and fourth position respectively. In general,

**Table 2. Top 10 artists by number of covered songs (weighted indegree).**

| Artist | Active period | Songs | Main genre | Style | Weighted indegree | Weighted outdegree |
|---|---|---|---|---|---|---|
| The Beatles | 1960s | 2096 | Pop/Rock | British Invasion | 16825 | 112 |
| Duke Ellington | 1920s–1970s | 7447 | Jazz | Big Band | 10564 | 262 |
| Bing Crosby | 1920s–1970s | 8864 | Vocal | Traditional Pop | 7011 | 676 |
| Bob Dylan | 1960s–2010s | 2378 | Pop/Rock | Folk/Country Rock | 5986 | 215 |
| Thelonious Monk | 1940s–1980s | 1526 | Jazz | BeBop | 4860 | 76 |
| Elvis Presley | 1950s–1970s | 5175 | Pop/Rock | Rock & Roll/Roots | 4692 | 516 |
| Frank Sinatra | 1930s–1990s | 8052 | Vocal | Traditional Pop | 4556 | 820 |
| Miles Davis | 1940s–1990s | 4323 | Jazz | BeBop | 3676 | 193 |
| Fred Astaire | 1920s–1980s | 1724 | Vocal | Show tunes | 3621 | 109 |
| Judy Garland | 1920s–1960s | 3278 | Vocal | Cast Recording | 3468 | 195 |

It includes number of songs from Allmusic and weighted outdegree.

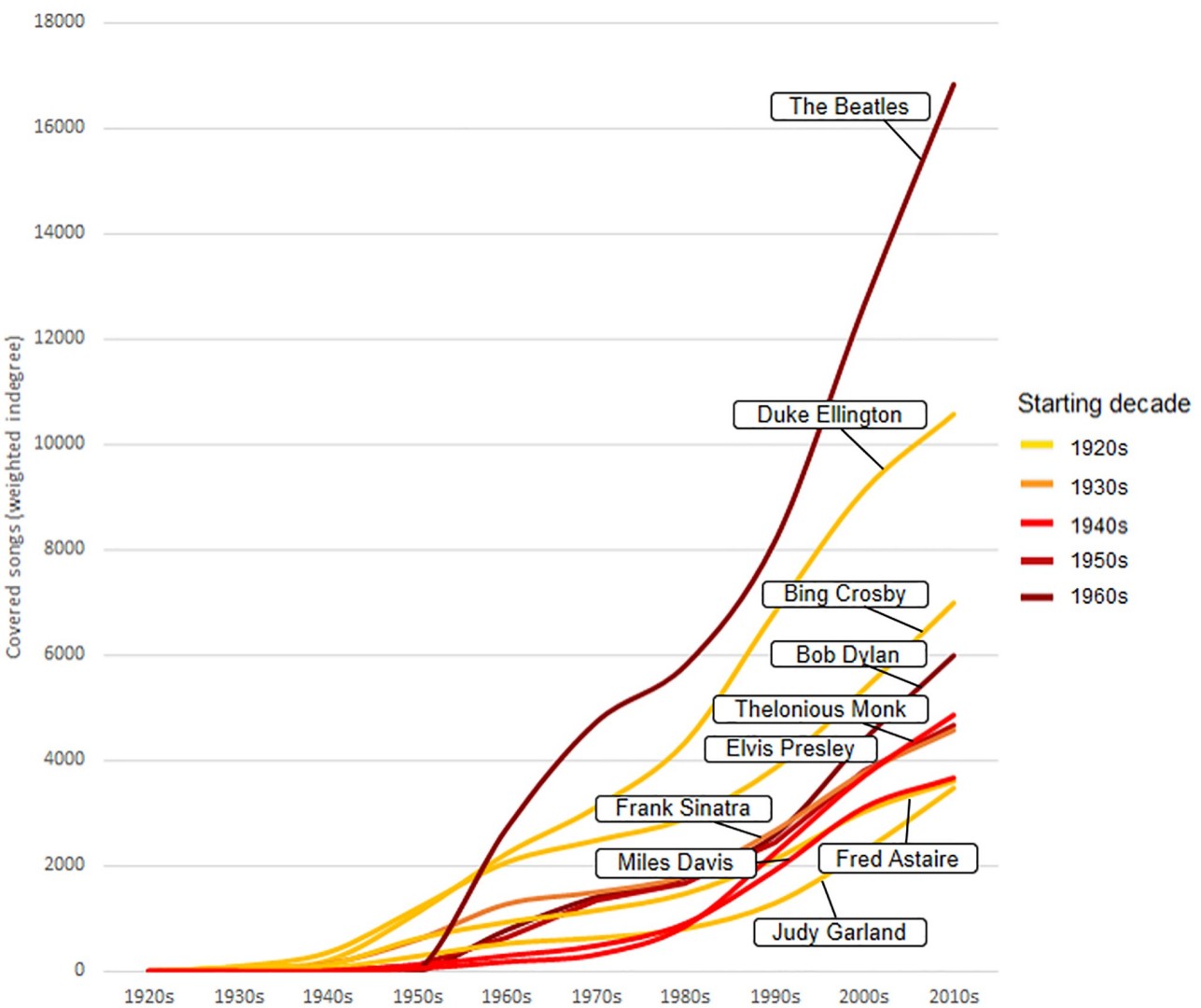

**Fig 4. Evolution of the number of covered songs (weighted indegree) of the first ten artists in each decade.**

this result suggests that as the cover network evolves over time, the impact of new figures is greater and faster. This would cause that the covering of songs is more and more concentrated in few artists, increasing extremely their impact over time. In addition, the sudden appearance and success of The Beatles and Bob Dylan would be representative of the generational change in the 1960s, as previously observed (Fig 3).

However, weighted indegree shows a cumulative impact that increases throughout the decades. A way to know the influence of artists in each moment is by calculating its PageRank. This metric shows the importance of an artist according to who their songs are covered by, rewarding covers from popular performers (who in turn are also very versioned) and under-valuing remakes from little-versioned musicians. Now, Bob Dylan climbs to the second position and new *Pop/Rock* classical figures, such as David Bowie, The Rolling Stones and Chuck Berry, emerge (Table 3). Other pioneers such as Prince's Band, an original jazz band and author of some of the most important Ragtime jazz standards, and Ethel Merman, a central figure of 1930s' musical theatre, acquire more relevance.

**Table 3. Top 10 artists by PageRank.**

| Artist | Active period | Songs | Main genre | Style | PageRank |
|--------|---------------|-------|------------|-------|----------|
| The Beatles | 1960s–1970s | 2096 | Pop/Rock | British Invasion | 0.00887 |
| Bob Dylan | 1960s–2010s | 2378 | Pop/Rock | Folk/Country Rock | 0.004559 |
| Bing Crosby | 1920s–1970s | 8864 | Vocal | Traditional Pop | 0.003301 |
| Duke Ellington | 1920s–1970s | 7447 | Jazz | Big Band | 0.002846 |
| David Bowie | 1960s–2010s | 1715 | Pop/Rock | Hard Rock | 0.002725 |
| The Rolling Stones | 1960s–2010s | 1359 | Pop/Rock | Hard Rock | 0.002607 |
| Prince's Band | 1900s–1920s | 7 | Jazz | Ragtime | 0.002574 |
| Chuck Berry | 1950s–2010s | 1222 | Pop/Rock | Rock & Roll/Roots | 0.002556 |
| The Velvet Underground | 1960s–1990s | 386 | Pop/Rock | Experimental Rock | 0.002339 |
| Ethel Merman | 1930s–1980s | 698 | Vocal | Traditional Pop | 0.002236 |

Fig 5 illustrates with more detail how the influence of these top artists occurs in different periods. We observe thus that the Prince's Band acquires great prominence in the initial period of the 1900s-1920s, being a key figure in the development of the early jazz. Then, some figures of the jazz (Duke Ellington) and vocal music (Bing Crosby and Ethel Merman) dominate the network in the 1920s-1950s, coinciding with the height of the Jazz Age. However, since the 1950s, all these artists experience a decline in favour of the new Pop/Rock stars. These new performers (The Beatles, Bob Dylan, The Rolling Stones), raised from the 1960s, increase their importance throughout the following decades until today.

Another way to select important artists on the music scene is through weighted betweenness centrality, which measures musicians' tendency to cover songs from distant and different artists, either by genre, language or epoch, and indicates the artist's degree of crossover. However, this metric cannot be interpreted according to the direction of the arc, making it impossible to know whether the position of an artist is due to the cover received or made [33]. To aid interpretation, the weighted indegree and weighted outdegree are also considered (Table 4). The first positions of the ranking (1st–6th) are occupied by performers who are covered far more frequently than they cover, which suggests that these artists (Frank Sinatra, Bing Crosby, Bob Dylan, The Beatles) are bridges that influence later performers from a broad range of spheres. Conversely, as recognized composers and arrangers of numerous popular hits, James Last and Ray Conniff covered more songs than they were covered. They are, then, bridges that are influenced by classical artists from different fields. Acker Bilk follows the same behaviour because he was a representative of Trad Jazz, a revival of Dixieland and New Orleans Jazz.

## Genres and styles

This section attempts an in-depth exploration of the musical impact of adaptations between genres over time, by observing when new trends emerged and what their main influences were.

Fig 6 depicts the full network of artists (nodes) connected by covers versions (arcs). Supplementary material S1 Fig includes a full-size version of the Fig 6. The colour of the artists represents their main genre, the size of nodes the number of recorded songs and the label size the number of covers received. Due to the great size of Pop/Rock genre, it was subdivided in styles (Rock & Roll/Roots, Pop/Rock, British invasion, Soft Rock, Psychedelic/Garage, Hard Rock, Country Rock, Foreign Language Rock, Art-Rock/Experimental, Punk/New Wave, Heavy Metal and Alternative/Indie Rock) to observe groups and connections between them. A first impression reveals that the artists tend to be grouped by genre, resulting in compact clusters such as *Jazz*, with 73% of internal covers, *Country*, with 43%, or *Blues*, with 37%. This high

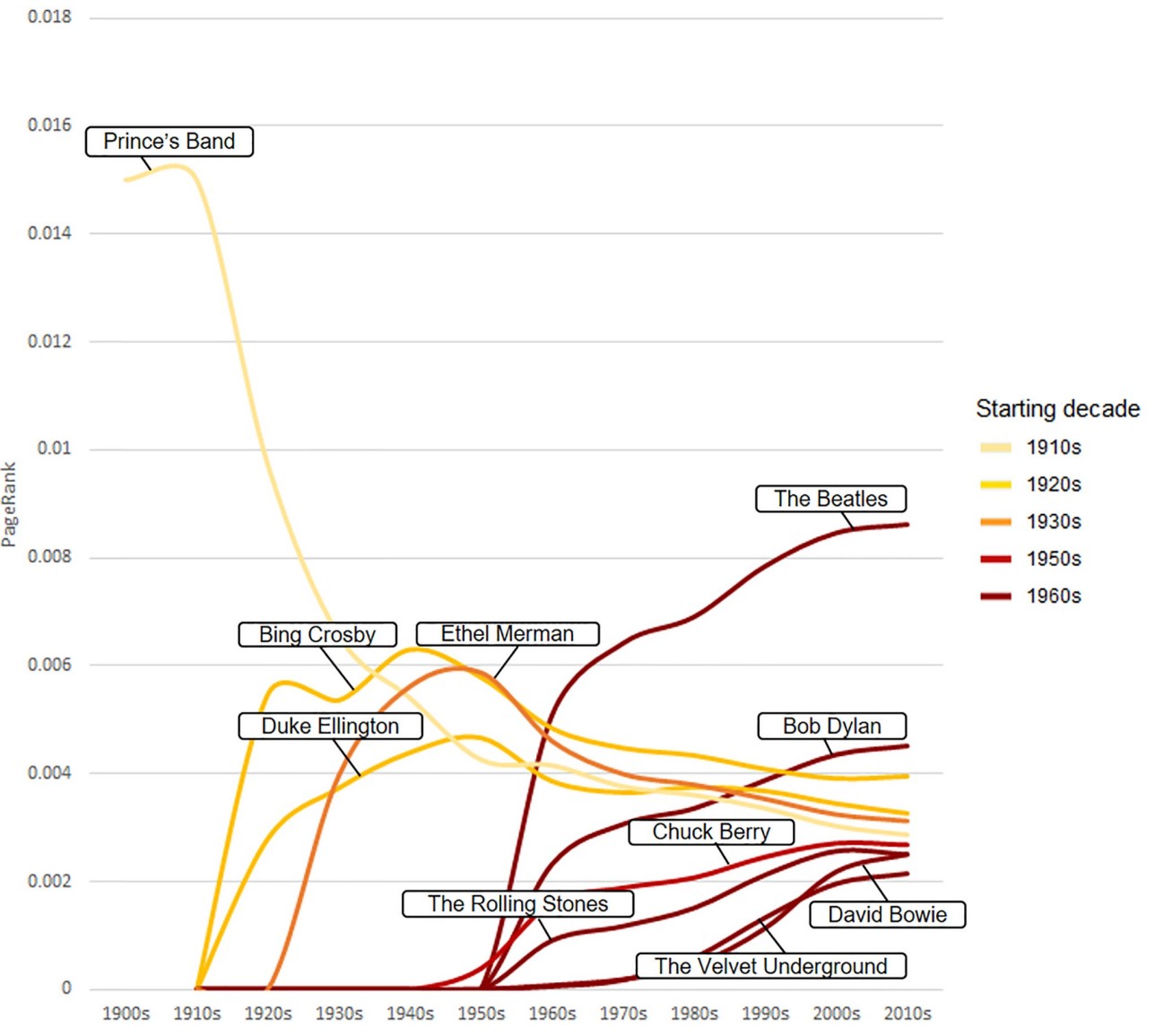

**Fig 5. Evolution of the PageRank of the first ten artists in each decade.**

cohesion would indicate conservative genres with long traditions and few external influences. Closely connected genres are also visible: *Jazz* and *Vocal*, with a 24% of Vocal singers covered by Jazz musicians and 20% in reverse. The reason could be that these genres were very popular from the 1910s–1940s, when many Jazz bands (e.g. The Dorsey Brothers, Duke Ellington's Orchestra) included singers or crooners (e.g. Bing Crosby, Louis Armstrong) who interpreted their pieces [34]. Less perceptible is the influence of language on the connection between artists due mainly to the predominance of English-speaking musicians (63%). The lower right section features a compact cluster of Brazilian musicians around Antonio Carlos Jobim and a less dense cluster of French singers near to Edith Piaf and Serge Gainsbourg.

We created a genre network connected by the number of covers between them (Fig 7). The arc weights were normalized per each node, and any weight below 0.1 was removed. This is the same as only including arcs where at least 10% of all the cover songs performed

**Table 4. Top 10 artists by weighted betweenness centrality.**

| Artist | Active period | Songs | Main genre | Style | Weighted indegree | Weighted outdegree | Weighted Betweenness centrality |
|---|---|---|---|---|---|---|---|
| Frank Sinatra | 1930s–1990s | 8052 | Vocal | Traditional Pop | 4556 | 820 | 348112988 |
| Bing Crosby | 1920s–1970s | 8864 | Vocal | Traditional Pop | 7011 | 676 | 224973559 |
| Bob Dylan | 1960s–2010s | 2378 | Pop/Rock | Folk/Country Rock | 5986 | 215 | 211717237 |
| The Beatles | 1960s–1970s | 2096 | Pop/Rock | British Invasion | 16825 | 112 | 132659702 |
| Elvis Presley | 1950s–1970s | 5175 | Pop/Rock | Rock & Roll/Roots | 4692 | 516 | 124172981 |
| Johnny Cash | 1950s–2000s | 4946 | Country | Country Gospel | 1224 | 468 | 98411237 |
| Acker Bilk | 1950s–2000s | 1202 | Jazz | Trad Jazz | 120 | 282 | 39169103 |
| Ray Conniff | 1930s–1990s | 1805 | Easy Listening | Orchestral/Easy Listening | 503 | 755 | 33349589 |
| David Bowie | 1960s–2010s | 1715 | Pop/Rock | Hard Rock | 2360 | 91 | 32278473 |
| James Last | 1960s–2010s | 2757 | Easy Listening | Mood Music | 79 | 453 | 21715877 |

It also includes weighted indegree, weighted outdegree and total songs.

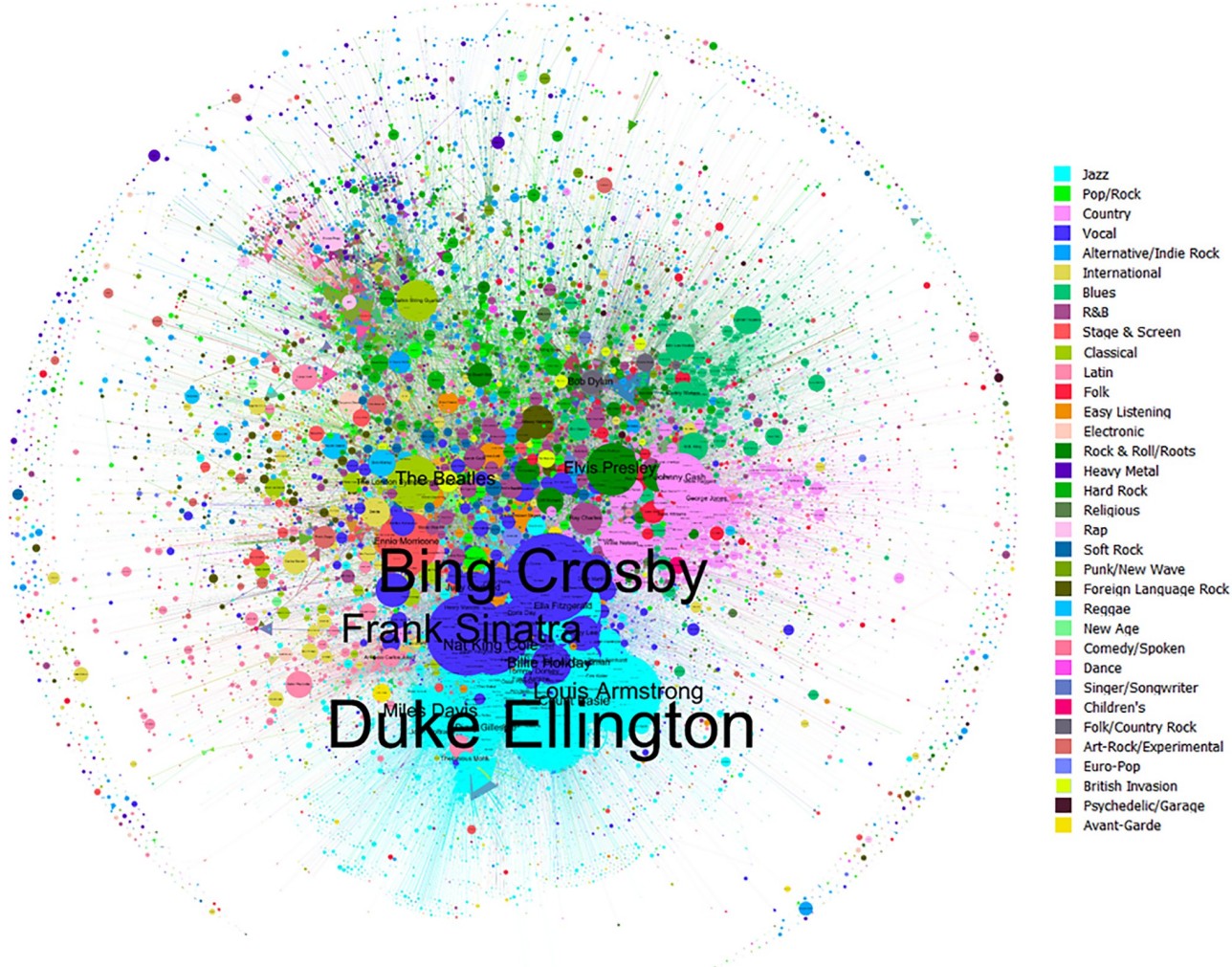

**Fig 6. Directed network graph of artists (N = 106,095) linked through covers (N = 855,650), coloured according to music genre, with node size according to number of recorded songs (Layout = Fruchterman Reingold).**

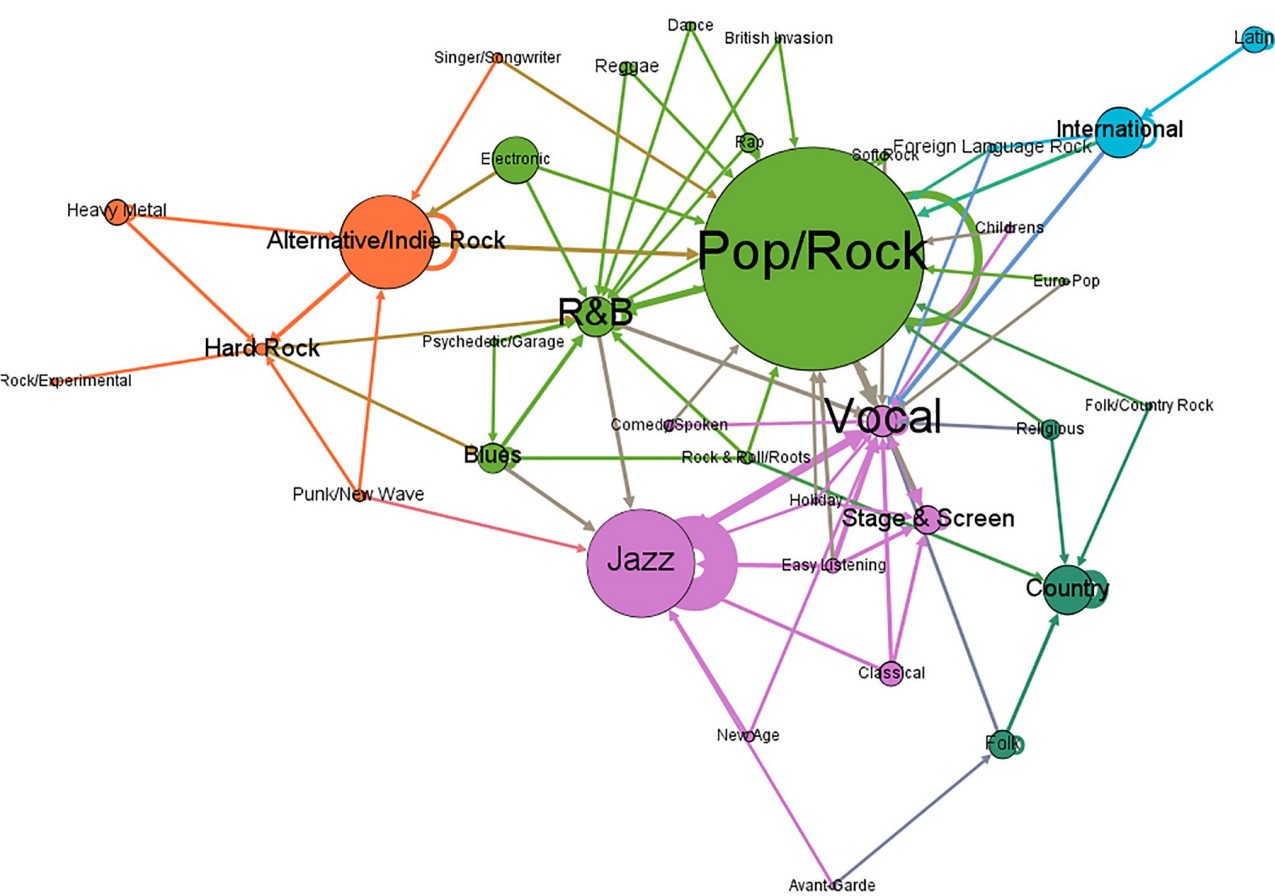

**Fig 7. Cover flows between music genres (N = 35) (arcs≥10% of outdegree) (Layout = Yifan Hu).**

(outdegree). For example, *Jazz* adapts 40% of *Jazz* pieces and 24% of *Vocal* songs and is an inspiration for 65% of *Jazz* and 10% of *Vocal* songs. This pruning criterion allows to select only the most important relationships among genres. Based on these relationships, five modules were identified according to the Louvain method (Resolution = 0.75, modularity = 0.279). The lower half of the picture includes all pre-Pop/Rock genres; early urban music (i.e. *Jazz*, *Vocal*, *Stage & Screen*, etc.) is shown in violet; and rural genres, headed by *Country*, in dark green. The upper half gathers the post-Pop/Rock genres. The core of *Pop/Rock* is highlighted in green and linked to the genres directly associated with it: *R&B* (Rhythm and Blues) and *Blues* and contemporary rock led by *Alternative/Indie Rock*. The blue cluster groups music genres from languages and cultures other than English (i.e. *Latin*, *International* and *Foreign Language Rock*), again revealing a language pattern in versioned songs. Thus, the genre network seems to indicate that there are two clear groups of genres: a cluster of old styles headed by *Jazz* and *Vocal*, and all the genres resulting from the advent of *Pop/Rock*.

Fig 8 depicts the proportion of cover songs in each decade grouped by the music genre of the covered artists. For instead, artists from the 1900s cover 55% of songs belonged to Vocal artists and 17% to Classical authors. Whereas artists appeared in the 2010s prefer to cover songs from Pop/Rock (39%) and Vocal (11.7%) artists. Notice that Pop/Rock styles have been merged to reduce lines and make clear the growth of the Pop/Rock music. This picture allows to appreciate how the music preferences change over time and the impact of new genres. Thus, the image shows how the Vocal and Jazz music describe high covering percentages until the

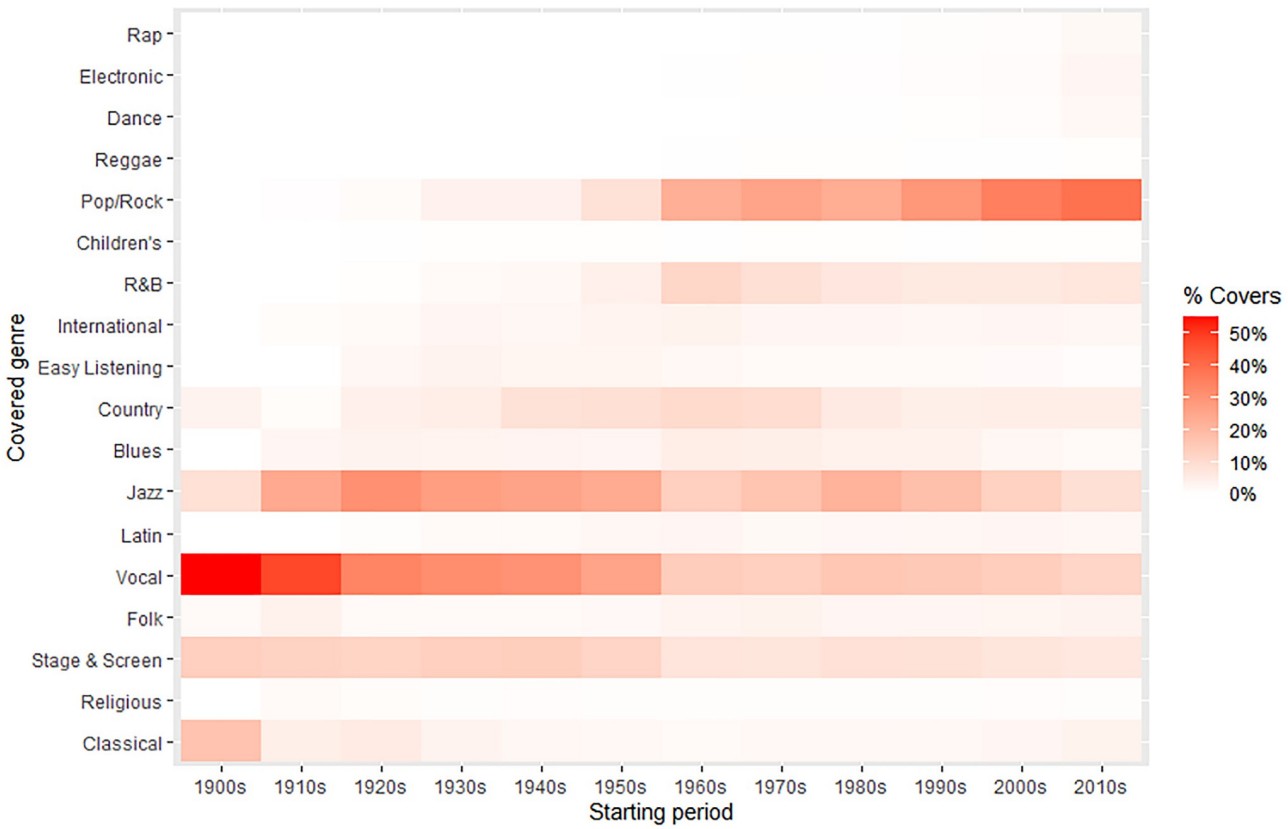

**Fig 8. Heat map with the percentage of covered music genres by decade.**

1960s, the moment in which Pop/Rock and their styles strongly emerge to dominate the music scene until today. It is interesting to notice the high peak of cover versions of R&B artists in the 1960s, which could indicate the great adaptation of black music by white performers through the Rock and roll and British rhythm and blues [35, 36].

## Discussion

The use of song adaptations between artists as an impact measure has made it possible to observe how the influence and appreciation of artists are distributed and how this has affected the creation and evolution of music genres. Firstly, the versioning of songs is a very unbalanced phenomenon, in which a small percentage of artists (less than 1%) are repeatedly versioned, while the vast majority (78%) are not. This unbalanced environment could be due to the low density ($d<0.01\%$) and poor frequency of cover songs (9.8 covers per artist against 79 songs per artist), which would determine that musicians prioritise to cover the most popular and prestigious artists [20].

Perhaps, one of the most interesting results regarding the evolution of the whole network is that the average of covers (weighted average degree) does not constantly grow, but since the 1950s decade, it gradually descends. This inflection point suggests an important transformation in that time, in which post-1950s generations of musicians stop doing remakes of previous songs. This perception is reinforced when cover versions between generations are observed (Fig 2). Then, it is possible to appreciate that in the 1910s-1950s period, artists have a predilection for covering songs from 1920s musicians, but this trend is broken since the 1950s, when

new performers prefer to cover songs from 1960s artists. These two marked periods coincide with the raising of the Jazz Age (1920s) and the Rock Revolution (1950s), which suggests that the appearance of these music genres would imply something more than just different tastes [18, 37, 38]. Both music genres could be consequence of deep transformations in the way of making music, including their perception on cover songs. During the 1900s-1950s period, when jazz bands and vocal singers dominated the music scene, the covering of songs was more frequent because they would compete in the performance of popular hits. The frequent use of standards in Jazz could reflect a greater predilection for the interpretation and improvisation [39]. However, with the advent of the Pop/Rock in the 1950s, the cover song lost prominence because musicians compete now in originality and creativity, considering cover versions as an appropriation [21, 35, 36]. In this context, cover songs tend to be used as a tribute to significant artists that serve as inspiration [40, 41].

This evolution in the use and meaning of cover versions can be also appreciate in the top covered artists. Both weighted indegree (Fig 3) and PageRank (Fig 4) have showed that musicians from the pre-Pop/Rock era (e.g. Duke Ellington, Bing Crosby) accumulate a constant number of covers, while subsequent performers from the 1960s (e.g. The Beatles, Bob Dylan) quickly overtake the number of covers of the previous generations. Taking into account that the covering of songs descends since 1950s, the fast success of these figures could illustrate an extreme concentration of the impact in special artists thereafter. PageRank, that avoids this cumulative effect, describes with more detail how Jazz and Vocal performers slowly lose their influence from the 1950s in favour of the new Pop/Rock stars, who rapidly increase their influence from the 1960s onwards. Furthermore, these network metrics have demonstrated great utility for describing different types of impacts such as cumulative (weighted indegree), influence (PageRank) and crossover (weighted betweenness), improving our appreciation on the role of different artists in the evolution of popular music.

The network graph visualization reports that genre affinity is the main criteria for covering songs among musicians, secondary reasons being language and nationality (e.g. Brazilian *bossa nova*, French *chanson*). As we have previously discussed, *Jazz* and *Pop/Rock* form the axes on which contemporary popular music lies [18, 37, 38]. The genres network shows that *Jazz* and *Vocal* are the most influential genres for pre-Pop/Rock styles, while *Pop/Rock* is the touchstone for subsequent genres. From a longitudinal perspective, the percentage of cover versions by genre in different decades (Fig 8) evidenced again how pre-Pop/Rock genres, mainly Jazz and Vocal, are the most covered genres until the 1950s. The moment in which Pop/Rock and associated styles grow up to be the most dominant genre at the beginning of the 21$^{st}$ Century.

## Limitations

However, the results of this study are limited to the sources used. Although the site Second-HandSongs collects most covers, it is not always completely comprehensive and certain minority artists and local styles (i.e. *flamenco*, *raï*, *afrobeat*) may not be well represented. Genre classification is according to Allmusic.com and some categories, such as the arc between Pop and Rock, are debatable. These limitations can influence the results relative to the relationship among genders.

Another point of discussion could be the normalization of covers by the number of recorded songs because one could argue that as more songs are released by an artist more chance to be covered. However, these normalizations in skewed distributions could produce some artefacts, in which one-hit wonders would accumulate a disproportionate impact with regard to the real influence of these artists (see an example in Supplementary materials

S2 File). An added problem is that the number of total songs in Allmusic is equivalent to the total number of different recordings, even when these different sessions are about the same song. This introduces a bias against the most popular singers, because they have more published material than others (i.e. greatest hits, live discs, hidden sessions) and their relative impact could be reduced.

## Conclusions

The obtained results let us to conclude that the cover versions network experiences a strong growth of remakes during the first decades until the 1950s, and from that moment the average of cover versions by artist gradually descends. This trend has led to confirm the existence of generational differences when the musicians cover songs. These patterns have been able be explained by the emergence of Jazz in the 1920s and the Pop/Rock in the 1950s.

The use of different network metrics has allowed to rank the most important artists in each moment, identifying the great impact of important musical figures such as Duke Ellington, Bing Crosby, The Beatles and Bob Dylan and their role in the height of the Jazz and the arrival of the Pop/Rock. These metrics have described different types of impact, providing significant nuances in the assessment of musicians.

Finally, cover versions network has shown that the affinity among genres and styles is the main criteria to cover songs. The genre network has illustrated how Jazz music and Pop/Rock shape the core of the popular music in the 20th and 21st centuries. Jazz constitutes the axis of old genres such as Vocal and Stage & Screen, whereas Pop/Rock is central for Rock styles such as R&B and Alternative/Indie Rock. This analysis at the genre level has also made possible to appreciate the change in the preferences for covering songs, confirming the transition of the Jazz Age in the 1920s to the Rock Revolution in the 1950s.

## Supporting information

**S1 File. Full artists indicators.**
(CSV)

**S2 File.**
(DOCX)

**S1 Fig.**
(TIF)

## Author Contributions

**Conceptualization:** José Luis Ortega.

**Data curation:** José Luis Ortega.

**Formal analysis:** José Luis Ortega.

**Funding acquisition:** José Luis Ortega.

**Investigation:** José Luis Ortega.

**Methodology:** José Luis Ortega.

**Project administration:** José Luis Ortega.

**Resources:** José Luis Ortega.

**Software:** José Luis Ortega.

**Supervision:** José Luis Ortega.

**Validation:** José Luis Ortega.

**Visualization:** José Luis Ortega.

**Writing – original draft:** José Luis Ortega.

**Writing – review & editing:** José Luis Ortega.

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
