## [Decision Letter · Decision Letter 0]

6 Nov 2020

PONE-D-20-26285

Cover versions as an impact indicator in popular music: a quantitative network analysis

PLOS ONE

Dear Dr. Ortega,

Thank you for submitting your manuscript to PLOS ONE. After careful consideration, we feel that it has merit but does not fully meet PLOS ONE’s publication criteria as it currently stands. Therefore, we invite you to submit a revised version of the manuscript that addresses the points raised during the review process.

We look forward to receiving your revised manuscript.

Kind regards,

Nicola Perra

Academic Editor

PLOS ONE

Journal Requirements:

Additional Editor Comments (if provided):

Dear authors,

as clear from the reviewers comments the work has merit and deserve to be published.

Please take into considerations all the comments raised by both reviewers. I believe that addressing them would really help and strengthen the paper.

Best regards,

Nicola Perra

Reviewers' comments:

Reviewer's Responses to Questions

**Comments to the Author**

1. Is the manuscript technically sound, and do the data support the conclusions?

Reviewer #1: Partly

Reviewer #2: Yes

2. Has the statistical analysis been performed appropriately and rigorously? 

Reviewer #1: No

Reviewer #2: Yes

3. Have the authors made all data underlying the findings in their manuscript fully available?

Reviewer #1: Yes

Reviewer #2: No

4. Is the manuscript presented in an intelligible fashion and written in standard English?

Reviewer #1: Yes

Reviewer #2: Yes

5. Review Comments to the Author

Reviewer #1: The manuscript “Cover versions as an impact indicator in popular music: A quantitative network analysis” uses a large crowdsourced database of musical covers (the performance of an already existing song by a different artist) to look at the network of covers. The analysis extracts the most central artists according to three measures of centrality. It also examines the connections between genres of covers. Finally, it provides some temporal analysis of how covers have evolved over the last century.

I think that the question of how covers result in social and musical connectivity is interesting and that networks are an appropriate method for parsing this phenomenon. However, I am not sure that the analysis in the manuscript has used networks to their fullest extent, and many of the analyses seem to be confounded by time. The question of time, as in how long an artist has existed, seems to be central to many of the analyses---it is the motivation for introducing PageRank, and at the center of many of the genre analyses. However, for how much it comes up, I do not think it has been directly and adequately addressed.

For example, the analysis observes that ranking artists by in-degree results in a number of older bands. It then states “a more precise way to reduce the oldness effect on the musical impact is by calculating its PageRank.” I do not think that PageRank is a sufficient answer here---it does explicitly not do anything to guarantee that time is properly accounted for in the analysis. A number of the artists in Table 2 are still from the 1920s and 1930s (brief aside: I would like to see more than the top 10 artists, perhaps as an appendix table). Instead, I think time should be tackled directly: we are interested in who the most central artists are in the cover network, but an artist is more likely to be covered simply if they are around longer. A simple analysis might be normalizing the in-degree by the number of years since the artist first became active. A more comprehensive analysis, that I would like to see, would treat the cover network as a temporal network and use temporal centrality measures to understand how artists have become more or less central over time. The individual artists section culminates in the statement that “musical oldness of artists is a key factor to improving the likelihood of being covered.” This seems more like the analysis hasn’t properly accounted for time, rather than a result. That is why I would like to see time handled more directly.

The genre analysis does a bit more to look at temporal dynamics, but it also avoids probing too deeply. I think Figure 6 is interesting. I think that it would work better as a heatmap where the rows and columns are years, and the color is Figure 6’s y-axis. I would like to know more about what is going on here. What genres are driving these temporal lags in covers? How does the centrality of genre covers change as a result in these lags?

I think that the manuscript struggles to pinpoint particular temporal questions to answer because its research questions are relatively broad and the main motivation is to do a quantitative analysis. I do not think doing a quantitative analysis necessarily tells us anything new, and so I don’t find the “total absence of quantitative studies” to be that compelling (though isn’t that also contradicted by the previous two paragraphs in literature review?). However, the manuscript mentions that qualitative studies have led to theorizing about the cultural contribution of covers (references 20-22). I think there is a missed opportunity here to use the large-scale network analysis to help clarify and expand upon what has been noticed in the theoretical/qualitative studies. I think by engaging more closely with the prior literature and synthesizing it into hypotheses that have not been tested would help generate a more productive analysis, and help guide questions about how time affects cover centrality.

Overall, I would like to see the issue of time addressed more clearly and more directly in the analysis. I think it confounds how we interpret both the individual and genre analyses. To this end and deriving questions related to how time affects the cover network, I think the manuscript would benefit from engaging with the hypotheses of the prior literature, and building from them, rather than doing a general descriptive analysis of the network.

Other comments

- The introduction should tell me what the questions are and how the analysis proposes to answer them. It ends abruptly before going to the related research. Similarly, the manuscript says that the objective is “to visualize and analyze the cover songs network between artists in order to describe its topology and identify the main nodes.” But why do that? What do we learn from making a network and identifying the main nodes that we didn’t know before? The final conclusion is “These results further strengthen the quantitative approach in studies on the history of music.” But what do they strengthen? What are they speaking to?

- I am worried about duplications in the data. In data cleaning, it is mentioned that if an artist covers the same song more than once, it is counted as a different cover. Has there been any validation to ensure that those covers of the same song are not exactly the same cover? similarly, I am worried that the number of songs in the Tables are extremely high. Duke Ellington was prolific, but I think it is unlikely that he wrote (or performed?) 7,447 songs. Similarly for the other artists, though there is then a huge gap with Franz Gruber and Joseph Mohr. I would like ot know what is going on here better. Why are the numbers so large?

- These analyses do not establish that the cover network is scale free and I think that the language of “scale free” should be left out of the paper. The degree distributions do not look scale free to me visually, and no test has been reported to indicate that they follow a power law. What is alpha? Relatedly, the discussion of “preferential attachment” should be dropped because no analysis establishes that there is preferential attachment in the network (exactly because there are no analyses that directly model the time of artists and receiving covers). I think that an analysis of preferential attachment would be interesting, and it can be done (for example, see Laszlo Barabasi’s textbook for measuring preferential attachment), but it is not done here. Even if it is done, I think the language of “scale free” should not be included---preferential attachment in a real world system does not guarantee a power law degree distribution.

- How PageRank is framed through the paper is inconsistent. In the introduction and abstract, it is described as measuring the “most influential.” In the analysis, it is described as accounting for time (which I don’t think it does). The intuitive description of PageRank when it is introduced as a metric is correct, and that should be used to describe in the other parts of the paper.

- I don’t believe there is a sentence that explicitly says what the nodes and edges are in the network. Specifically, I want to know if the nodes are artists from SecondHandSongs, or from Allmusic.

- In the SecondHandSongs dataset, what does it mean for a song to be labelled as “traditional.” In the Allmusic dataset, what is meant by “genres and styles are not interconnected”? Isn’t a network made of them later?

- The SecondHandSongs dataset section first says that there are 96 thousand songs. The data extraction paragraph says that 155 thousand songs were extracted. Why are they not the same number?

- Details about song and artist IDs do not need to be included in the manuscript.

- I do not understand the inclusion criteria for the genre network: “Only the most important influences were highlighted through percentages of covers equal to or higher than 10% the total number of covers made.”

- The names of artists and bands should not be italicized.

Reviewer #2: The manuscript ‘Cover versions as an impact indicator in popular music: a quantitative network analysis’ deals with the analysis of a database of cover songs through the lens of networks, where two artists are linked if one covers a song of the other. The resulting network is then used to measure the influence between artists and to exploit the connections between their respective genres.

The idea is (to the best of my knowledge) new, the results are interesting, and the presentation clear (the figures not so much). Overall, I believe that the manuscript deserves publication in PLoS One, but there are some points that I would like the authors to discuss first:

1) As far as I understand the network representation adopted by the authors, there is a directed link between artist i and artist j if i covers a song of j. The authors also write "Links between artists are therefore established according to the number of times an artist covers another", so this makes me think that links are weighted. However, all the measures used do not take weights into account. This is an important point that needs to be clarified. In case the measures take weights into account this should be explained better. Otherwise, why not considering weighted measures (in-/out-strength instead of degree, betweenness centrality defined on the shortest paths that consider weights etc.)?

2) Is there a way to account for the size of the production of different artists to make the cover flows comparable? For example, an artist with a huge production that gets covered a lot would could in principle be comparable with an artist with a smaller production that gets covered a bit less. The same argument holds for genre-to-genre comparisons, since the production in each year and for each genre can change. I believe that counters should be somehow normalised to be comparable. See for example what is usually done for citation networks [Radicchi, F., & Castellano, C. (2011). Rescaling citations of publications in physics. Physical Review E, 83(4), 046116.] or [Waltman, Ludo. "A review of the literature on citation impact indicators." Journal of informetrics 10.2 (2016): 365-391.] to assess the influence of authors and temporal flows of knowledge across different fields [Sun, Y., Latora, V. The evolution of knowledge within and across fields in modern physics. Sci Rep 10, 12097 (2020)].

Below a list of minor points:

1) Abstract: the sentence "However, no studies have explored the meaning of covers from a quantitative and structural perspective" sounds a bit vague to me. What is the meaning of a cover? I would suggest rephrasing it. (the paper is not about the meaning of covers as far as I see it). Also, the structural perspective might be misleading here, since it's not on the structure of the cover, but covers are used to create a structure between artists.

2) Introduction: "These acknowledgements lead to (...) while assessing the impact of artists’ careers". As this sentence is written at the moment is seems that covers access the impact of the careers. It should be rephrased.

3) Introduction, last sentence: "who their songs were versioned by" sounds a bit weird.

4) Related research, third paragraph: "Nevertheless, an important gap in the analysis of cover songs is the total absence of quantitative studies." The second paragraph presents some quantitative studies on music, not on covers. Given that the third paragraph starts with 'nevertheless', I would swap the writing to emphasise the lack of analyses of covers among the quantitative studies rather than the lack of quantitative studies among the analyses of covers.

5) Objectives. The second point reads "Could social network measures be used to detect different types of impact?". I understand this is precisely what the manuscript is about, but I do not think that the results of the paper actually answer this question. The manuscript does not compare the ranking induced by the measures with some "ground truth" influence of artists, since it obviously does not exist (maybe some proxy for popularity could be used). Thus, since the authors are not testing the ability of the measures to capture the influence, but are proposing to use the measures as a proxy for influence, I would change/remove the question.

6) A clear definition of the network representation and nomenclature is missing. This would help the reading of the following 'social network analysis' section, where nodes and lines are mentioned (also, why not calling them links?).

7) Degree distribution section: the \\alpha is not defined. Also, which method is used to fit the distribution? This should be specified. The fit does not seem to capture well the tail of the out degree distribution.

8) The authors focus on the Page Rank and the betweenness centrality, but there is obviously a long list of additional measures that could be used instead to produce rankings in -evolving- networks (Liao, Hao, et al. "Ranking in evolving complex networks." Physics Reports 689 (2017): 1-54.). For example, the alpha-centrality could be a good proxy for influence in static graphs. Why did the authors restricted their attention to the selected measures?

9) Just before Genres and styles section: "This result confirms that the musical oldness of artists is a key factor to improving the likelihood of being covered". This makes sense, but isn't there already a bias towards the "old" artists that survived against the new ones? There would be "old" forgotten ones that did not even appear in the database, so oldness in this sense already encodes an inner survivorship bias. The authors could maybe expand on this.

10) Fig 3 is not very readable.

11) I really like the idea behind Fig. 4. Would it be possible to visualise it also with a heatmap instead of a network? It would make possible to directly compare flows within and across genres. Ideally, it would be interesting to see how these flow evolved in time and reorganise due to the appearance of new genres.

12) Fig. 5 and Fig. 6 do not have labels.

13) Fig. 6 is not easily to interpret, since lines typically denote an evolution, while here each point is a percentage. Also, I noticed something that I do not understand. As I understand the chart, there are songs from artists of the 90s that are covered by artists that started in the 1910...(the lightest blue peak at 1990). How long can a career be? This should be checked, and maybe a simple stacked bar chart would facilitate the visualisation? Otherwise, I would suggest to use a heatmap with the years of the two artists on the axes and the proportion of covers as color intensity. This would make the comparison easier.

14) In the conclusions the authors state that "Most interestingly, perhaps, these influences are the result of cycles." The connection between influence and cycles is not clear to me and could be further explained.

15) Here some related papers that might be of interest:

- Serrà J, Corral Á, Boguñá M, Haro M, Arcos JL (2012) Measuring the evolution of contemporary western popular music.Sci. Rep.2:521.

- Zivic PHR, Shifres F, Cecchi GA (2013) Perceptual basis of evolving western musical styles. Proc. Natl. Acad. Sci. U.S.A.110(24):10034–10038.

- Youngblood, M. (2019). Conformity bias in the cultural transmission of music sampling traditions. Royal Society open science, 6(9), 191149.

6. PLOS authors have the option to publish the peer review history of their article (what does this mean?). If published, this will include your full peer review and any attached files.

Reviewer #1: No

Reviewer #2: **Yes: **Iacopo Iacopini

---

## [Author Response · Author response to Decision Letter 0]

16 Dec 2020

5. Review Comments to the Author

First of all, I would like to thank to the reviewers their comments because their suggestions have led me to detect important gaps in the manuscript and explore the evolutionary aspect of the data with more detail. Great part of the manuscript was then rewritten to centre the manuscript in how the network has evolved in successive decades.

Reviewer #1: The manuscript “Cover versions as an impact indicator in popular music: A quantitative network analysis” uses a large crowdsourced database of musical covers (the performance of an already existing song by a different artist) to look at the network of covers. The analysis extracts the most central artists according to three measures of centrality. It also examines the connections between genres of covers. Finally, it provides some temporal analysis of how covers have evolved over the last century.

I think that the question of how covers result in social and musical connectivity is interesting and that networks are an appropriate method for parsing this phenomenon. However, I am not sure that the analysis in the manuscript has used networks to their fullest extent, and many of the analyses seem to be confounded by time. The question of time, as in how long an artist has existed, seems to be central to many of the analyses---it is the motivation for introducing PageRank, and at the center of many of the genre analyses. However, for how much it comes up, I do not think it has been directly and adequately addressed.

For example, the analysis observes that ranking artists by in-degree results in a number of older bands. It then states “a more precise way to reduce the oldness effect on the musical impact is by calculating its PageRank.” I do not think that PageRank is a sufficient answer here---it does explicitly not do anything to guarantee that time is properly accounted for in the analysis. A number of the artists in Table 2 are still from the 1920s and 1930s (brief aside: I would like to see more than the top 10 artists, perhaps as an appendix table). Instead, I think time should be tackled directly: we are interested in who the most central artists are in the cover network, but an artist is more likely to be covered simply if they are around longer. A simple analysis might be normalizing the in-degree by the number of years since the artist first became active. A more comprehensive analysis, that I would like to see, would treat the cover network as a temporal network and use temporal centrality measures to understand how artists have become more or less central over time. The individual artists section culminates in the statement that “musical oldness of artists is a key factor to improving the likelihood of being covered.” This seems more like the analysis hasn’t properly accounted for time, rather than a result. That is why I would like to see time handled more directly.

Section “Degree distributions” has been rename and broaden as “General characteristics of the covers network”. Now, the global network has been split by decade in which the artists start their music career, covering since 1910s to 2010s. This allows us to study how the network has evolved as new artist were aggregated. Apart from the indegee and outdegree distributions (Figure 1), this section now presents several global network indicators (Figure 2 and Table 1) such as Weighted average degree and Clustering coefficient that describe the characteristics of the network in each decade. This has allowed to observe different growing patterns. In addition, Figure 3 (former Figure 6) has been added to explain the possible causes of the irregular growth of the network.

A table with the network indicators of the artists and a large-size version of Fig 6 was added as supplementary material.

The genre analysis does a bit more to look at temporal dynamics, but it also avoids probing too deeply. I think Figure 6 is interesting. I think that it would work better as a heatmap where the rows and columns are years, and the color is Figure 6’s y-axis. I would like to know more about what is going on here. What genres are driving these temporal lags in covers? How does the centrality of genre covers change as a result in these lags?

Figure 6 (Now Figure 3) has been redone as a heatmap. Now, differences between generations are better appreciated. Section about genres has been remade to emphasize how the preferences covering genres have changed (Figure 8). This result attempts to explain the structural changes observed in the growth of the global network. 

I think that the manuscript struggles to pinpoint particular temporal questions to answer because its research questions are relatively broad and the main motivation is to do a quantitative analysis. I do not think doing a quantitative analysis necessarily tells us anything new, and so I don’t find the “total absence of quantitative studies” to be that compelling (though isn’t that also contradicted by the previous two paragraphs in literature review?). However, the manuscript mentions that qualitative studies have led to theorizing about the cultural contribution of covers (references 20-22). I think there is a missed opportunity here to use the large-scale network analysis to help clarify and expand upon what has been noticed in the theoretical/qualitative studies. I think by engaging more closely with the prior literature and synthesizing it into hypotheses that have not been tested would help generate a more productive analysis, and help guide questions about how time affects cover centrality.

The objectives of the paper have been changed due to the new approach of the manuscript and all the objectives include now an evolutionary perspective.

Overall, I would like to see the issue of time addressed more clearly and more directly in the analysis. I think it confounds how we interpret both the individual and genre analyses. To this end and deriving questions related to how time affects the cover network, I think the manuscript would benefit from engaging with the hypotheses of the prior literature, and building from them, rather than doing a general descriptive analysis of the network.

Results section has been enriched with longitudinal analyses that respond to the research questions. Now, the discussion about these results has been improved adding more previous studies that support the interpretations 

Other comments

- The introduction should tell me what the questions are and how the analysis proposes to answer them. It ends abruptly before going to the related research. Similarly, the manuscript says that the objective is “to visualize and analyze the cover songs network between artists in order to describe its topology and identify the main nodes.” But why do that? What do we learn from making a network and identifying the main nodes that we didn’t know before? The final conclusion is “These results further strengthen the quantitative approach in studies on the history of music.” But what do they strengthen? What are they speaking to?

A paragraph was added at the end of the introduction to define the principal purpose of the paper. As I have commented, objectives have been changed and now more accurate research queries were formulated. Conclusions section was added to give more relevance to this part and new conclusions were written to precise the main findings.

- I am worried about duplications in the data. In data cleaning, it is mentioned that if an artist covers the same song more than once, it is counted as a different cover. Has there been any validation to ensure that those covers of the same song are not exactly the same cover? 

This only occurs with songs that share the same title. It is very infrequent that an artist covers a same song from the same artist in different moments. In spite of this, I think that each cover versions shall be counted as different because it is created in a different context and they express the strength in the connection between artists. 

similarly, I am worried that the number of songs in the Tables are extremely high. Duke Ellington was prolific, but I think it is unlikely that he wrote (or performed?) 7,447 songs. Similarly for the other artists, though there is then a huge gap with Franz Gruber and Joseph Mohr. I would like ot know what is going on here better. Why are the numbers so large?

The number of songs are obtained from Allmusic.org page. This site gathers all the songs recorded by an artist. This link shows the current number of Duke Ellington songs, 7621.

https://www.allmusic.com/artist/duke-ellington-mn0000120323/songs/all

This amount is only used to express the size of the node in the graph and they are used to illustrate the production of musicians. They are not relevant in the analysis. 

However, in my opinion that volume of recorded songs is common in important music starts. For example, Bing Crosby has close to 9000 songs

https://www.allmusic.com/artist/bing-crosby-mn0000094252/songs/all

The Beatles, 2000 songs

https://www.allmusic.com/artist/the-beatles-mn0000754032/songs/all

- These analyses do not establish that the cover network is scale free and I think that the language of “scale free” should be left out of the paper. The degree distributions do not look scale free to me visually, and no test has been reported to indicate that they follow a power law. What is alpha? Relatedly, the discussion of “preferential attachment” should be dropped because no analysis establishes that there is preferential attachment in the network (exactly because there are no analyses that directly model the time of artists and receiving covers). I think that an analysis of preferential attachment would be interesting, and it can be done (for example, see Laszlo Barabasi’s textbook for measuring preferential attachment), but it is not done here. Even if it is done, I think the language of “scale free” should not be included---preferential attachment in a real world system does not guarantee a power law degree distribution.

Allusions to scale free have been avoided. Due to the new perspective of the paper, focused on the evolutionary aspects, any issue on the type of network is irrelevant and it has been removed. 

Alpha is the exponent of the curve. This has been included in the text (p. 8)

- How PageRank is framed through the paper is inconsistent. In the introduction and abstract, it is described as measuring the “most influential.” In the analysis, it is described as accounting for time (which I don’t think it does). The intuitive description of PageRank when it is introduced as a metric is correct, and that should be used to describe in the other parts of the paper.

That reference about the use of Page Rank for counteracting time effects is wrong and was removed. This metrics is now only used as influence indicator. 

- I don’t believe there is a sentence that explicitly says what the nodes and edges are in the network. Specifically, I want to know if the nodes are artists from SecondHandSongs, or from Allmusic.

This sentence was added to General characteristics of the covers network in p. 8: “A network graph was constructed with 106,096 nodes (artists) and 855,650 arcs (cover versions) extracted from SecondHandSongs”

- In the SecondHandSongs dataset, what does it mean for a song to be labelled as “traditional.” In the Allmusic dataset, what is meant by “genres and styles are not interconnected”? Isn’t a network made of them later?

Traditional means that the first performer of the song is unknown, so then it is not possible to create a link between covered artist and subsequent performers.

That sentence about the interconnection of genres and styles is a mistake and that idea is wrongly expressed. A new text was added: “Styles are not considered as sub-categories of genres, then they are used as labels or keywords that are freely assigned to an artist.”

- The SecondHandSongs dataset section first says that there are 96 thousand songs. The data extraction paragraph says that 155 thousand songs were extracted. Why are they not the same number?

There is a mistake in the count of original songs. The sample only include titles of songs that have been covered, so 155k is actually the list of titles included in the sample, both covers and original songs. I have decided to remove that figure because it would be confused. 

- Details about song and artist IDs do not need to be included in the manuscript.

I think that this information is very important to reproduce the sample. This detail about the IDs allows other researchers to extract data from SecondHandSongs and Allmusic. 

- I do not understand the inclusion criteria for the genre network: “Only the most important influences were highlighted through percentages of covers equal to or higher than 10% the total number of covers made.”

This is a pruning process that selects only the most important links. The reason is that all the genres have links among them, resulting a ball-shape network. This pruning criterion ensures that only the most significant relationships are drawn in the picture, producing a more interesting graph. This text was added in p. 18: “The graph only includes links that are equal or higher than 10% of all the cover songs of other genres (outdegree). For example, Jazz adapts 40% of Jazz pieces and 24% of Vocal songs and is an inspiration for 65% of Jazz and 10% of Vocal songs. This pruning criterion allows to select only the most important relationships among genres.” 

- The names of artists and bands should not be italicized.

Corrected

Reviewer #2: The manuscript ‘Cover versions as an impact indicator in popular music: a quantitative network analysis’ deals with the analysis of a database of cover songs through the lens of networks, where two artists are linked if one covers a song of the other. The resulting network is then used to measure the influence between artists and to exploit the connections between their respective genres.

The idea is (to the best of my knowledge) new, the results are interesting, and the presentation clear (the figures not so much). Overall, I believe that the manuscript deserves publication in PLoS One, but there are some points that I would like the authors to discuss first:

1) As far as I understand the network representation adopted by the authors, there is a directed link between artist i and artist j if i covers a song of j. The authors also write "Links between artists are therefore established according to the number of times an artist covers another", so this makes me think that links are weighted. However, all the measures used do not take weights into account. This is an important point that needs to be clarified. In case the measures take weights into account this should be explained better. Otherwise, why not considering weighted measures (in-/out-strength instead of degree, betweenness centrality defined on the shortest paths that consider weights etc.)?

I agree with you, and it was a mistake to use those metrics for a weighted network. Degree was replaced by the Weighted degree. Now, the results are more consistent. 

2) Is there a way to account for the size of the production of different artists to make the cover flows comparable? For example, an artist with a huge production that gets covered a lot would could in principle be comparable with an artist with a smaller production that gets covered a bit less. The same argument holds for genre-to-genre comparisons, since the production in each year and for each genre can change. I believe that counters should be somehow normalised to be comparable. See for example what is usually done for citation networks [Radicchi, F., & Castellano, C. (2011). Rescaling citations of publications in physics. Physical Review E, 83(4), 046116.] or [Waltman, Ludo. "A review of the literature on citation impact indicators." Journal of informetrics 10.2 (2016): 365-391.] to assess the influence of authors and temporal flows of knowledge across different fields [Sun, Y., Latora, V. The evolution of knowledge within and across fields in modern physics. Sci Rep 10, 12097 (2020)].

This is an interesting point because one could assume that some type of normalization is necessary to avoid size-effects. However, this issue is problematic from a conceptual perspective. Firstly, we are talking about musicians, not about scientists. The concept of production (number of papers vs. number of songs) and impact (number of citations and number of covered songs) cannot be literally translated. In Science, any paper includes bibliographic references (it is almost impossible to publish a research article without references), then as more articles more citations. In other words, citations are dependent of the publications.

However, songs do not include any reference to previous releases, musicians are not determined to cover songs of other artists. Therefore, more songs do not imply more cover versions. In fact, there are musicians with a large trajectory and very low number of cover versions, and the contrary (e.g. tribute bands). This is confirmed by the low correlation between the number of songs recorded and covers done (weighted outdegree) (r=.465) and received (weighted indegree) (r=.443). Then, the use of a normalized count would cause numerical artefacts, undervaluing great musical figures with an extensive career and large production (e.g. Duke Ellington, Bing Crosby) and overvaluing unknown artists whose exclusive merit is to firstly interpret a popular song such as Silent night by Franz Gruber and Joseph Mohr, or Summertime by Abbie Mitchell. In my opinion, the use of normalized counts would cause confusing results. 

Below a list of minor points:

1) Abstract: the sentence "However, no studies have explored the meaning of covers from a quantitative and structural perspective" sounds a bit vague to me. What is the meaning of a cover? I would suggest rephrasing it. (the paper is not about the meaning of covers as far as I see it). Also, the structural perspective might be misleading here, since it's not on the structure of the cover, but covers are used to create a structure between artists.

Abstract was rewritten and that sentence was removed due to its ambiguity. Now, the abstract is focused on the new results and their conclusions.

2) Introduction: "These acknowledgements lead to (...) while assessing the impact of artists’ careers". As this sentence is written at the moment is seems that covers access the impact of the careers. It should be rephrased.

The sentence was rewritten: “These acknowledgements lead to links between artists and illustrate the musical influences of soloists or bands, while enabling to assess the impact of artists on specific musical communities.”

3) Introduction, last sentence: "who their songs were versioned by" sounds a bit weird.

The sentence has been rewritten: “…including information about cover songs and their performers.”

4) Related research, third paragraph: "Nevertheless, an important gap in the analysis of cover songs is the total absence of quantitative studies." The second paragraph presents some quantitative studies on music, not on covers. Given that the third paragraph starts with 'nevertheless', I would swap the writing to emphasise the lack of analyses of covers among the quantitative studies rather than the lack of quantitative studies among the analyses of covers.

I agree with your observation. The text was rephrased: “Nevertheless, an important gap in the quantitative studies on the contemporary popular music is the total absence of analysis on cover versions. In most of the cases, they have been observed from a theoretical perspective, figuring out the cultural meaning of this type of songs.”

5) Objectives. The second point reads "Could social network measures be used to detect different types of impact?". I understand this is precisely what the manuscript is about, but I do not think that the results of the paper actually answer this question. The manuscript does not compare the ranking induced by the measures with some "ground truth" influence of artists, since it obviously does not exist (maybe some proxy for popularity could be used). Thus, since the authors are not testing the ability of the measures to capture the influence, but are proposing to use the measures as a proxy for influence, I would change/remove the question.

That research question was reformulated: “Which musical artists are the most influential in the network and how have they reached that position? What information does each network metric provide about the impact?”

6) A clear definition of the network representation and nomenclature is missing. This would help the reading of the following 'social network analysis' section, where nodes and lines are mentioned (also, why not calling them links?).

This information is now included at the beginning of the General characteristics of the covers network.

7) Degree distribution section: the \\alpha is not defined. Also, which method is used to fit the distribution? This should be specified. The fit does not seem to capture well the tail of the out degree distribution.

This text was added to explain the power-law distributions: “This interpretation could be confirmed by the cumulative distribution of covers performed (outdegree) and received (indegree) by artist. They follow a power-law distribution (Figure 1), with an exponent (α) of 2.882 for outdegree and 2.937 for indegree. R package poweRlaw was used to create the cumulative distribution functions (CDF) and the maximum likelihood estimation (MLE) method was used to fit the distributions.”

8) The authors focus on the Page Rank and the betweenness centrality, but there is obviously a long list of additional measures that could be used instead to produce rankings in -evolving- networks (Liao, Hao, et al. "Ranking in evolving complex networks." Physics Reports 689 (2017): 1-54.). For example, the alpha-centrality could be a good proxy for influence in static graphs. Why did the authors restricted their attention to the selected measures?

The study is focused on three metrics: Weighted indegree, PageRank and Betweeeness centrality. These metrics provide three different points of view about the impact: Weighted indegree offers information about how the impact is accumulated, PageRank about the influence capacity of artists, and the betweenness centrality measures the ability to be covered by different and distant artists. These metrics were selected by several reasons: (1) they are well known by a generalist audience (PLOS ONE is a multidisciplinary journal), (2) their meaning and calculation is easy to understand, then it is easy to translate their meaning to a different environment such as the popular music, and (3) they are implemented in many network analysis software packages (Gephi, Pajek, VOSviewer, etc.).

I agree that there much more network metrics, but this is a study about cover songs and I think that three metrics is enough to have a good view about the impact of the musicians. I think that the inclusion of more metrics would not improve the results nor provide additional information. For example, alpha-centrality belongs to the eigenvector family (PageRank is a version of eigenvector for directed networks). It is very likely that the result would be similar than the observed with PageRank.

9) Just before Genres and styles section: "This result confirms that the musical oldness of artists is a key factor to improving the likelihood of being covered". This makes sense, but isn't there already a bias towards the "old" artists that survived against the new ones? There would be "old" forgotten ones that did not even appear in the database, so oldness in this sense already encodes an inner survivorship bias. The authors could maybe expand on this.

That paragraph was removed due to that result was obvious and to make way to other relevant results. 

10) Fig 3 is not very readable.

Figure 3 (now Figure 6) was changed by a more complete figure. The problem of this figure is the size and resolution. I think that the journal would publish a large-size version of the figure that allows to explore the connections and groups.

11) I really like the idea behind Fig. 4. Would it be possible to visualise it also with a heatmap instead of a network? It would make possible to directly compare flows within and across genres. Ideally, it would be interesting to see how these flow evolved in time and reorganise due to the appearance of new genres.

Figure 8 was added to visualize the evolution of cover songs by music genre. This figure illustrates the preference by certain genre in each decade.

12) Fig. 5 and Fig. 6 do not have labels.

Both figures has been removed

13) Fig. 6 is not easily to interpret, since lines typically denote an evolution, while here each point is a percentage. Also, I noticed something that I do not understand. As I understand the chart, there are songs from artists of the 90s that are covered by artists that started in the 1910...(the lightest blue peak at 1990). How long can a career be? This should be checked, and maybe a simple stacked bar chart would facilitate the visualisation? Otherwise, I would suggest to use a heatmap with the years of the two artists on the axes and the proportion of covers as color intensity. This would make the comparison easier.

Figure 6 was change by Figure 3. I agree with you and I think that a heatmap is more suitable to represent that idea.

You comment about artists from the 1910s cover songs from 1990s artists. This is a small artefact caused by the low proportion of songs and artists from 1900s decade in the database. The reason of that weird result is that there are some old orchestras and choruses (e.g. The London Symphony Orchestra) that have a large trajectory and they sometimes cover songs from Pop/Rock stars. 

14) In the conclusions the authors state that "Most interestingly, perhaps, these influences are the result of cycles." The connection between influence and cycles is not clear to me and could be further explained.

Conclusions were completely rewritten and that sentence was removed. The new analysis makes more clear the connection between cycles and influence.

15) Here some related papers that might be of interest:

- Serrà J, Corral Á, Boguñá M, Haro M, Arcos JL (2012) Measuring the evolution of contemporary western popular music.Sci. Rep.2:521.

- Zivic PHR, Shifres F, Cecchi GA (2013) Perceptual basis of evolving western musical styles. Proc. Natl. Acad. Sci. U.S.A.110(24):10034–10038.

- Youngblood, M. (2019). Conformity bias in the cultural transmission of music sampling traditions. Royal Society open science, 6(9), 191149.

Many thanks for the references. I have decided to cite the last reference (Youngblood, 2019) because this study analyses music samples (a bit close to the cover versions) and it is focused on the factors that influence their selection. The remaining ones have been discarded because they are centered on musicology aspects such as the sound, melody and rhythm.

---

## [Decision Letter · Decision Letter 1]

3 Feb 2021

PONE-D-20-26285R1

Cover versions as an impact indicator in popular music: a quantitative network analysis

PLOS ONE

Dear Dr. Ortega,

Thank you for submitting your manuscript to PLOS ONE. After careful consideration, we feel that it has merit but does not fully meet PLOS ONE’s publication criteria as it currently stands. Therefore, we invite you to submit a revised version of the manuscript that addresses the points raised during the review process.

ACADEMIC EDITOR: Dear Authors, as acknowledged by the two reviewers the article has been drastically improved in the first revision. Please consider the minor points raised by the referees and resubmit. In doing so, please make sure to highlight the changes to the manuscript so that we can speed up the review process.

We look forward to receiving your revised manuscript.

Kind regards,

Nicola Perra

Academic Editor

PLOS ONE

Reviewers' comments:

Reviewer's Responses to Questions

**Comments to the Author**

1. If the authors have adequately addressed your comments raised in a previous round of review and you feel that this manuscript is now acceptable for publication, you may indicate that here to bypass the “Comments to the Author” section, enter your conflict of interest statement in the “Confidential to Editor” section, and submit your "Accept" recommendation.

Reviewer #1: (No Response)

Reviewer #2: (No Response)

2. Is the manuscript technically sound, and do the data support the conclusions?

Reviewer #1: Partly

Reviewer #2: Yes

3. Has the statistical analysis been performed appropriately and rigorously? 

Reviewer #1: Yes

Reviewer #2: N/A

4. Have the authors made all data underlying the findings in their manuscript fully available?

Reviewer #1: Yes

Reviewer #2: Yes

5. Is the manuscript presented in an intelligible fashion and written in standard English?

Reviewer #1: Yes

Reviewer #2: Yes

6. Review Comments to the Author

Reviewer #1: The manuscript “Cover versions as an impact indicator in popular music: a quantitative network analysis” has improved with the revisions. I think that identifying the split in what decades are covered (1920s vs 1950s) is interesting, and I think it has helped create a more coherent story around the other analyses. I have some reservations and questions that I would still like to see addressed.

1. Table 1 and several of the figures rely on networks constructed across time. How exactly were those networks made? Is it a cumulative network, where, for example, any node and link in the 1910s network is also in the 1920s network, and any node and link in the 1910s or 1920s network is in the 1930s network? I think this is the proper way to construct the network for the analyses. But it’s not clear if instead these might be snapshot networks, where only nodes from a particular decade and who they cover are included. This should be explained in the text.

2. I am still skeptical of duplicates in the songs used to count the productivity of each artist, as seen in Table 2 for example. Looking even just at one artist, Duke Ellington, it was easy to see that several “different” songs were actually the same one. I suspect it is similar for the other artists. If this measure of productivity is used in the paper, it should do deduplication.

Also, it is not clear how artists were matched across SecondHandSongs and Allmusic. Is it just if an artist’s name matches exactly across the two datasets? This should be explained.

3. I am still unclear on what nodes ultimately end up in the networks because there are several similar but conflicting numbers about the number of nodes

- In Data Sources, it says the site “reported 788k covers and, 143k artists”

- The paper reports that “143k artists and 1.1 million covers” were extracted. This contradicts the first number. Is it because the previous stats were reported at an earlier time? If so, I think they should be dropped because they are confusing and appear to contradict the data collection.

- IDs 0 to 175000 were extracted. Why 175000 if there are only 143k artists?

- The network is reported to have 106,095 nodes. Why is it not 143k?

- The last column of Table 1 reports that there are 101,112 nodes in the 2010s network. Why is it not 106,095?

If there are data processing steps that are causing these different numbers, they should be clearly reported next to each step.

4. Figures 4 and 5 help make some of the findings clear. It is difficult to tell which curve is associated with each artist. I would suggest placing each artist’s name at the end of each curve, so that they are all aligned on the right side of the plot.

Figure 8 is interesting, but it is very difficult to tell which color lines up with which curve. I think a heatmap-style figure like Figure 3 would be more clear.

5. There are several places in the paper where it says it will do an analysis that is more than what is actually done

- In Top Covered Artists, the paper says “This section analyzes who are these figures and how they have acquired their positions.” The analysis does not say anything about *how* they acquired their positions in the network, just their importance in it.

- In Top Covered Artists, the paper says “In general, as the cover network evolves over time, the impact of new figures is greater and faster.” This has not been shown in general. In the following sentence, it also hasn’t been shown that anything is specifically “exponential.”

- In the Discussion, the paper says “Regardless of these limitations, the results could be representative of mainstream popular music.” That this is representative is not clear to me, I don’t think the paper needs to say this.

I particularly would like to see the methods questions answered. Addressing the other points would help make the paper’s story more clear and reliable.

-------

Other comments:

- The paper’s language switches tenses often. Sometimes it’s past tense, sometimes it’s present, sometimes it’s conditional. A reliable tense should be used throughout the paper.

- Network metrics (“weighted in-degree”, “clustering coefficient”) should not be capitalized throughout the paper.

- The abstract is jarring: it jumps right into explaining data after barely motivating why the structure of covers is interesting (and it is interesting).

- The density by itself does not tell us much about the centralization of the network. Many empirical networks have very low density. What is more interesting is the density over time. When the density is reported, I think the “less than” sign is backwards, it should be (d < 0.01%) if I understand correctly. Also is it that d < 0.01% or 0.01? A density of 0.0001 for 0.01% does seem very low.

- It still wasn’t clear to me immediately how the edges of the genre network were selected. If I understand correctly, it’s more clear to say something like “The edge weights were normalized per each node, and any weight below 0.1 was removed. This is the same as only including links where at least 10%...” For future reference, this is similar to but less statistically rigorous than using a backboning method like the disparity filter.

- In the discussion, it’s not clear if the “deep transformations in the way of making music” are speculations, or already known by previous work. How can “subsequent performers from the 1960s quickly overtake the number of covers of the previous generations” if “the cover song lost prominence because musicians compete now”?

-------

Typos and phrasing (since PLoS ONE doesn’t provide copy editing):

- Page 1, “network metrics have allowed to identify” is missing a word I think

- Page 2, “while enabling to assess the impact” is awkwardly phrased

- Page 4, “In this form, it attempts to know how the use of” is awkward with “it attempts to know.” The paper can be more definitive.

- Page 5, In the third research question, “possible to appreciate” is awkward since “appreciate” is usually meant in terms of subjective terms.

- Page 5, “the database has been fed,” “fed” is an awkward verb. Databases don’t eat

- Page 5, “the song is labeled as traditional,” put quotes around “traditional” to make it more clear it’s a label, and say that it’s labeled that way by the dataset (I believe?) rather than by the researcher

- Page 6, “Styles are not considered as subcategories of genres, then they are used as labels”, something isn’t right with the second half, “then they are used as”

- Be consistent with “arcs,” “lines, “edges,” “links.” Aim to use just one

- Page 12, very bottom, typo with “in favour of the new Pop/Rock starts.” Should be “stars”

- Page 15, very bottom, “For instead” doesn’t seem like the right transition

Reviewer #2: I would like to that the author for considering the suggestions given by both referees (especially those regarding the temporal aspect). I find the current manuscript much improved. The are some points that I would like to discuss before recommending the manuscript for publication.

1) I am glad that the author modified the measures as suggested and took weights into account. In fact, answering a point previously raised, the author says that the "Degree was replaced by the Weighed degree. Now, the results are more consistent." Was it the same for the other measures? At the moment it is not clear to me if all measures have been changed to their weighted counterparts or just the degree. This is important for consistency and reproducibility.

2) Regarding my previous point (2), I obviously agree that musicians are not scientists and a song does not come with a set of references. However, artists with a huge productions have a lot of material available that could be used by other artists when performing covers. In very basic terms, let us assume that an artist wants to record a cover. In the absence of real preferences (tastes or "influence"), the artist would pick a random song from the ones available, and thus would have a higher probability of covering artists with a big production (due to the mass of the production and the age). This implies that --even in the absence of a real influence mechanism-- artists with a lot of songs will end up being more covered than others. I understand the concerns of the author, but, given the central role played by 'influence as a driver for covering' in this manuscript, I believe that this is an important point that has not been addressed and should be (at least) discussed.

In addition, I find the argument 'more songs do not imply more cover versions' not really confirmed here. The correlations coefficients provided "low correlation between the number of songs recorded and covers done (weighted outdegree) (r=.465) and received (weighted indegree) (r=.443)" are not that low to me.

Minor points:

When I read 'evolutionary' I think about evolution as a change (like in biology). I thus find its use in some parts of the manuscript a bit our of context, since it reads more like a temporal analysis (how quantities change in time) rather than the analysis of how things develop (by changing into others).

In Objectives, 'indicators' is repeated.

Just before Fig.2, 'The artists that joint the network' joined

The text regarding Fig. 2 says "From a longitudinal point of view, the growing pattern of the network is not continuous (Figure 2)". I find the sentence 'growing pattern of the network' a bit misleading, since degree distributions are skewed (and the average is not much representative) and the real pattern I assume would be hidden in the fact that new coming nodes without many links pull down the average after the initial growing phase. The author might consider plotting also the number of new nodes/covers in time to have a clearer picture of the growing pattern of the network that generate the curves shown. In addition, I would suggest avoiding the use of 'not continuous' since it might imply the presence of a discontinuity in the curve (which is not the case).

After Fig. 4, "This fact then suggests that the covering of songs is more and more concentrated in few artists, increasing exponentially their impact over time." how is "exponentially" tested?

In Fig. 4 and 5 I find difficult to associate artist labels with curves when there are many (for example The Rolling Stones and Bob Dylan in FIg.5). Maybe arrows could help?

Fig. 8 is hard to read since blues colors are only slightly different (I can guess due to their size, but it is not straightforward).

Figure captions seem to include caption titles only. These should be improved by better explaining the variables shown.

7. PLOS authors have the option to publish the peer review history of their article (what does this mean?). If published, this will include your full peer review and any attached files.

Reviewer #1: No

Reviewer #2: **Yes: **Iacopo Iacopini

---

## [Author Response · Author response to Decision Letter 1]

9 Feb 2021

Reviewer #1: The manuscript “Cover versions as an impact indicator in popular music: a quantitative network analysis” has improved with the revisions. I think that identifying the split in what decades are covered (1920s vs 1950s) is interesting, and I think it has helped create a more coherent story around the other analyses. I have some reservations and questions that I would still like to see addressed.

1. Table 1 and several of the figures rely on networks constructed across time. How exactly were those networks made? Is it a cumulative network, where, for example, any node and link in the 1910s network is also in the 1920s network, and any node and link in the 1910s or 1920s network is in the 1930s network? I think this is the proper way to construct the network for the analyses. But it’s not clear if instead these might be snapshot networks, where only nodes from a particular decade and who they cover are included. This should be explained in the text.

I have introduced some lines to explain that the longitudinal analysis is done over cumulative networks:

“In order to study the evolution of the network, artists were classified by the decade in which they started their musical career. During this process, 4,983 artists were discarded because they did not have that information. Then, several cumulative networks by decade were built. This aggregate approach is because the cover versions are made of current or precedent artists and each network rests on the previous one.”

2. I am still skeptical of duplicates in the songs used to count the productivity of each artist, as seen in Table 2 for example. Looking even just at one artist, Duke Ellington, it was easy to see that several “different” songs were actually the same one. I suspect it is similar for the other artists. If this measure of productivity is used in the paper, it should do deduplication.

The number of songs reported by Allmusic is not exactly the total number of different musical pieces interpreted or recorded. Rather, it is the total number of releases, including different recording of the same songs. Due to this problem, that metric is not used in the study as productivity measure and it is included only for illustrative purpose. Even then, I decided to include a footnote to explain the meaning of songs:

“Data supplied by Allmusic. Number of musical tracks recorded and released by artist. It may include different recordings of the same song.” 

Also, it is not clear how artists were matched across SecondHandSongs and Allmusic. Is it just if an artist’s name matches exactly across the two datasets? This should be explained.

A paragraph was added to Data cleaning section explaining how the match was made:

“The name of the artist or band was used to match SecondHandSongs and Allmusic records. This element was carefully cleaned to avoid mismatches between slight name variations. Thus, some characters such as ampersands and punctuation marks were removed or replaced by normal letters (Simon & Garfunkel by Simon and Garfunkel, “Big Mama” Thornton by Big Mama Thornton). In other cases, names with different variations were normalized (Booker T. and the Memphis Group’s by Booker T. and the M.G,’s, Johnny Burnette Trio by Johnny Burnette Rock and Roll Trio); synonymy between different artists (Tom Jones, 60s vocal artist and Tom Jones, 50s lyricist) was solved adding a digit to the name.”

3. I am still unclear on what nodes ultimately end up in the networks because there are several similar but conflicting numbers about the number of nodes

- In Data Sources, it says the site “reported 788k covers and, 143k artists”

- The paper reports that “143k artists and 1.1 million covers” were extracted. This contradicts the first number. Is it because the previous stats were reported at an earlier time? If so, I think they should be dropped because they are confusing and appear to contradict the data collection.

You are right those figures are contradictory, the total number of covers reported by the site in January 2020 should not be less than the total retrieved covers. The problem is that the “1.1 million covers” refers to an unpolished data sample, which also included originals, samples and medleys. I have removed that text and I have added a new paragraph in Data cleaning explaining all the transformations of the sample and the final number of covers and artists used in the study:

“When a song is traditional or the original performer is unknown, SecondHandSongs assigns it to a generic unknown (20,428). These covers were removed because there is not a connection between artists. In addition, SecondHandSongs joins the code of two or more artists when they perform together (e.g. Sinatra and Bono, Elvis Presley and the Jordanaires). These cases were duplicated, assigning the same song to each artist. After all these transformations the final sample included 106k artists and 855k cover songs.”

- IDs 0 to 175000 were extracted. Why 175000 if there are only 143k artists?

SecondhandSongs and Allmusic use sequential IDs. This has the problem that when a record is removed (i.e. duplicate, wrong name, etc.), the gap is not filled. Then, the total number of records in the database is always less than the last IDs. In addition, and due to this problem, it is necessary a large IDs edge, to be sure that there are no more records after some empty IDs. 

- The network is reported to have 106,095 nodes. Why is it not 143k?

This is explained in the new paragraph in Data cleaning.

- The last column of Table 1 reports that there are 101,112 nodes in the 2010s network. Why is it not 106,095?

This is due to 4,983 artists do not have starting period date. The reason is that they are not included in Allmusic or it does not provide that information for those artists. This line was added to the second paragraph of General characteristics of the covers network section:

“During this process, 4,983 artists were discarded because they did not have that information.”

If there are data processing steps that are causing these different numbers, they should be clearly reported next to each step.

4. Figures 4 and 5 help make some of the findings clear. It is difficult to tell which curve is associated with each artist. I would suggest placing each artist’s name at the end of each curve, so that they are all aligned on the right side of the plot.

I have decided to insert a fine line to connect each label with the trend. The idea of putting the label at the end is fine, but the problem is that some lines finish very close among them and there is no place for the labels.

Figure 8 is interesting, but it is very difficult to tell which color lines up with which curve. I think a heatmap-style figure like Figure 3 would be more clear.

Lines of the Figure 8 has been replaced by a heat map.

5. There are several places in the paper where it says it will do an analysis that is more than what is actually done

- In Top Covered Artists, the paper says “This section analyzes who are these figures and how they have acquired their positions.” The analysis does not say anything about *how* they acquired their positions in the network, just their importance in it.

The sentence has been rewritten: “This section analyses who are these figures and how they have evolved.”

- In Top Covered Artists, the paper says “In general, as the cover network evolves over time, the impact of new figures is greater and faster.” This has not been shown in general. 

The sentence has been rewritten: “In general, this result suggests that as the cover network evolves over time, the impact of new figures is greater and faster.”

In the following sentence, it also hasn’t been shown that anything is specifically “exponential.”

The sentence has been rewritten: “This would cause that the covering of songs is more and more concentrated in few artists, increasing extremely their impact over time.”

- In the Discussion, the paper says “Regardless of these limitations, the results could be representative of mainstream popular music.” That this is representative is not clear to me, I don’t think the paper needs to say this.

That sentence was removed and replaced by: “These limitations can influence the results relative to the relationship among genders.”

I particularly would like to see the methods questions answered. Addressing the other points would help make the paper’s story more clear and reliable.

-------

Other comments:

- The paper’s language switches tenses often. Sometimes it’s past tense, sometimes it’s present, sometimes it’s conditional. A reliable tense should be used throughout the paper.

Paper has been revised to uniform the tenses

- Network metrics (“weighted in-degree”, “clustering coefficient”) should not be capitalized throughout the paper.

Corrected

- The abstract is jarring: it jumps right into explaining data after barely motivating why the structure of covers is interesting (and it is interesting).

It is possible that the abstract quickly focuses on methods and results, but there a 250 words limitation and I think that it is more relevant to summarize the results than including sentences about the motivations of the study.

- The density by itself does not tell us much about the centralization of the network. Many empirical networks have very low density. What is more interesting is the density over time. When the density is reported, I think the “less than” sign is backwards, it should be (d < 0.01%) if I understand correctly. Also is it that d < 0.01% or 0.01? A density of 0.0001 for 0.01% does seem very low.

It was a typo error. Because density is the proportion of links over the total possible links, then it is common to be expressed in percentages. 

- It still wasn’t clear to me immediately how the edges of the genre network were selected. If I understand correctly, it’s more clear to say something like “The edge weights were normalized per each node, and any weight below 0.1 was removed. This is the same as only including links where at least 10%...” For future reference, this is similar to but less statistically rigorous than using a backboning method like the disparity filter.

Many thanks for the suggested text. I have included it in the paper. I used this criterion to prone the network because I did not find other way to do it. Thank you for your recommendation, I see that disparity filter has a R package, so I will try to use it in next studies.

- In the discussion, it’s not clear if the “deep transformations in the way of making music” are speculations, or already known by previous work. How can “subsequent performers from the 1960s quickly overtake the number of covers of the previous generations” if “the cover song lost prominence because musicians compete now”?

The exact meaning of the sentence is a speculation mine but based on the previous references. In general, academic literature agrees that the history of contemporary popular music in the 20th century is defined by two differentiate epochs: Jazz and Rock.

This sentence was added to explain those contradictory statements: “Taking into account that the covering of songs descends since 1950s, the fast success of these figures could illustrate an extreme concentration of the impact in special artists thereafter.”

-------

Typos and phrasing (since PLoS ONE doesn’t provide copy editing):

- Page 1, “network metrics have allowed to identify” is missing a word I think

- Page 2, “while enabling to assess the impact” is awkwardly phrased

The sentence was rewritten: “while assessing the impact of artists’ careers in specific musical communities.”

- Page 4, “In this form, it attempts to know how the use of” is awkward with “it attempts to know.” The paper can be more definitive.

The sentence was rewritten: “it attempts to understand how”

- Page 5, In the third research question, “possible to appreciate” is awkward since “appreciate” is usually meant in terms of subjective terms.

Appreciate has been replaced by observe

- Page 5, “the database has been fed,” “fed” is an awkward verb. Databases don’t eat

Fed has been replaced by supplied

- Page 5, “the song is labeled as traditional,” put quotes around “traditional” to make it more clear it’s a label, and say that it’s labeled that way by the dataset (I believe?) rather than by the researcher

That sentence was removed because it is better explained in Page 7

- Page 6, “Styles are not considered as subcategories of genres, then they are used as labels”, something isn’t right with the second half, “then they are used as”

The sentence was rewritten: “therefore, they are labels or keywords freely assigned to an artist.”

- Be consistent with “arcs,” “lines, “edges,” “links.” Aim to use just one

Corrected

- Page 12, very bottom, typo with “in favour of the new Pop/Rock starts.” Should be “stars”

Corrected

- Page 15, very bottom, “For instead” doesn’t seem like the right transition

For instead is used to introduce an example. I think it is fine. 

Reviewer #2: I would like to that the author for considering the suggestions given by both referees (especially those regarding the temporal aspect). I find the current manuscript much improved. The are some points that I would like to discuss before recommending the manuscript for publication.

1) I am glad that the author modified the measures as suggested and took weights into account. In fact, answering a point previously raised, the author says that the "Degree was replaced by the Weighed degree. Now, the results are more consistent." Was it the same for the other measures? At the moment it is not clear to me if all measures have been changed to their weighted counterparts or just the degree. This is important for consistency and reproducibility.

Betweenness centrality has been recalculated taking weights into account (see Table 4). This is also made explicit in the text, replacing betweenness centrality by weighted betweenness centrality.

From the first version, PageRank was calculated using weights. A sentence was added to Social network analysis > PageRank: “It was calculated taking weights into account.” 

2) Regarding my previous point (2), I obviously agree that musicians are not scientists and a song does not come with a set of references. However, artists with a huge productions have a lot of material available that could be used by other artists when performing covers. In very basic terms, let us assume that an artist wants to record a cover. In the absence of real preferences (tastes or "influence"), the artist would pick a random song from the ones available, and thus would have a higher probability of covering artists with a big production (due to the mass of the production and the age). This implies that --even in the absence of a real influence mechanism-- artists with a lot of songs will end up being more covered than others. I understand the concerns of the author, but, given the central role played by 'influence as a driver for covering' in this manuscript, I believe that this is an important point that has not been addressed and should be (at least) discussed.

In addition, I find the argument 'more songs do not imply more cover versions' not really confirmed here. The correlations coefficients provided "low correlation between the number of songs recorded and covers done (weighted outdegree) (r=.465) and received (weighted indegree) (r=.443)" are not that low to me.

I think that the discussion about the normalization of covers is very interesting but it goes beyond to the peer-review. Let me go in depth the reasons why the normalization is not suitable in this study:

As I have commented to the first reviewer, the information about the number of songs that Allmusic provides is not entirely correct. Allmusic computes the number of different recordings, even these different sessions are about the same song. This introduce a bias in favour of the most popular singers, because they have more published material than others (i.e. Greatest hits, Live discs, hidden sessions, etc.). I think that this variable is unstable and it is not suitable to normalize the covers.

Second, the normalization in skewed distributions can produce artefacts with unrealistic results. For example, if the normalization is applied, the top 10 artists will be:

Label songs active_period starting_decade main_genre weighted indegree nor weighted indegree

Franz Gruber 2 1890s 1890s Classical 3157 1578.5

Joseph Mohr 2 1890s 1890s Classical 3157 1578.5

Emily Laurey 1 1890s 1890s Classical 1557 1557

Meister Glee Singers 1 1890s 1890s Vocal 983 983

Hollace Shaw 1 1930s 1930s Stage & Screen 975 975

Hiram Sherman 1 1930s 1930s Vocal 966 966

Gloria Grafton 1 1930s 1930s Stage & Screen 944 944

Tally-Ho! 1 1900s 1900s Vocal 919 919

Kathryn Crawford 1 1930s 1930s Stage & Screen 770 770

The result shows that these authors have a very high impact due to by only one or two songs. The first three are due to a popular Christmas song. The other one are unknown singers with only a punctual popular hit. The question is then, are these authors the most important in the contemporary popular music? I do not think so. Then, the normalization in this case provide poor results.

You say: “In the absence of real preferences (tastes or "influence"), the artist would pick a random song from the ones available, and thus would have a higher probability of covering artists with a big production (due to the mass of the production and the age).”

Correct. A random selection means that every song has the same likelihood to be elected, then the distribution will be normal or Gaussian. But the distribution is skewed, then it should be a non-random selection criterion and there is not the same likelihood to be elected. That is, it does not matter how many songs you have if they are not enough goods or interesting, they will not be covered.

I have included a footnote to explain why this normalization was not applied:

“Data supplied by Allmusic. Number of musical tracks recorded and released by artist. It may include different recordings of the same song. Due to this and the skewed distribution of cover songs (Figure 1), this data was not used as normalization variable.”

Minor points:

When I read 'evolutionary' I think about evolution as a change (like in biology). I thus find its use in some parts of the manuscript a bit our of context, since it reads more like a temporal analysis (how quantities change in time) rather than the analysis of how things develop (by changing into others).

I have replaced evolutionary by longitudinal

In Objectives, 'indicators' is repeated.

Just before Fig.2, 'The artists that joint the network' joined

Corrected

The text regarding Fig. 2 says "From a longitudinal point of view, the growing pattern of the network is not continuous (Figure 2)". I find the sentence 'growing pattern of the network' a bit misleading, since degree distributions are skewed (and the average is not much representative) and the real pattern I assume would be hidden in the fact that new coming nodes without many links pull down the average after the initial growing phase. The author might consider plotting also the number of new nodes/covers in time to have a clearer picture of the growing pattern of the network that generate the curves shown. In addition, I would suggest avoiding the use of 'not continuous' since it might imply the presence of a discontinuity in the curve (which is not the case).

Table 1 publishes the number of nodes and arcs in each time, so the readers can see the difference between each period. It is normal to assume that there are new coming nodes with a low number of covered songs. This is because precedent artists scarcely cover songs from subsequent artists. However, I do not see that new nodes pull down the average because in spite the new nodes are not covered still, the former ones are indeed being covered by the new nodes, then the average can growth (as it does happen in the 1900-1950s period).

I have replaced continuous by constant.

After Fig. 4, "This fact then suggests that the covering of songs is more and more concentrated in few artists, increasing exponentially their impact over time." how is "exponentially" tested?

Exponentially was used as a descriptive adverb. The trend was not tested. I have replace exponentially by extremely.

In Fig. 4 and 5 I find difficult to associate artist labels with curves when there are many (for example The Rolling Stones and Bob Dylan in FIg.5). Maybe arrows could help?

Those figures were edited to introduce arrows between labels and lines.

Fig. 8 is hard to read since blues colors are only slightly different (I can guess due to their size, but it is not straightforward).

Figure 8 was replaced by a heat map

Figure captions seem to include caption titles only. These should be improved by better explaining the variables shown.

Some captions have been rewritten to detail information about the variables.

---

## [Decision Letter · Decision Letter 2]

25 Mar 2021

PONE-D-20-26285R2

Cover versions as an impact indicator in popular music: a quantitative network analysis

PLOS ONE

Dear Dr. Ortega,

Thank you for submitting your manuscript to PLOS ONE. After careful consideration, we feel that it has merit but does not fully meet PLOS ONE’s publication criteria as it currently stands. Therefore, we invite you to submit a revised version of the manuscript that addresses the points raised during the review process.

I would invite the authors to address the last important point raised by the reviewer number 2. 

Many thanks for all the hard work in revising the manuscript

We look forward to receiving your revised manuscript.

Kind regards,

Nicola Perra

Academic Editor

PLOS ONE

Journal Requirements:

Reviewers' comments:

Reviewer's Responses to Questions

**Comments to the Author**

1. If the authors have adequately addressed your comments raised in a previous round of review and you feel that this manuscript is now acceptable for publication, you may indicate that here to bypass the “Comments to the Author” section, enter your conflict of interest statement in the “Confidential to Editor” section, and submit your "Accept" recommendation.

Reviewer #1: All comments have been addressed

Reviewer #2: All comments have been addressed

2. Is the manuscript technically sound, and do the data support the conclusions?

Reviewer #1: Yes

Reviewer #2: Yes

3. Has the statistical analysis been performed appropriately and rigorously? 

Reviewer #1: Yes

Reviewer #2: Yes

4. Have the authors made all data underlying the findings in their manuscript fully available?

Reviewer #1: Yes

Reviewer #2: Yes

5. Is the manuscript presented in an intelligible fashion and written in standard English?

Reviewer #1: Yes

Reviewer #2: Yes

6. Review Comments to the Author

Reviewer #1: The author has addressed most of my concerns. I appreciate the work that was done to respond to both of the reviewers.

Reviewer #2: I would like to that the author again for implementing the suggested modifications, improving in this way the overall presentation of the manuscript. Based on the latest changes, I would recommend publication in PLoS One --even though I am still a bit puzzled by the entire normalisation argument.

In particular, there are two crucial points from the latest answer:

1) When the author writes "the information about the number of songs that Allmusic provides is not entirely correct. Allmusic computes the number of different recordings, even these different sessions are about the same song (...) I think that this variable is unstable and it is not suitable to normalize the covers." it seems to me that the non-suitability of the possessed data is used to justify the appropriateness of the employed methodology. While the possibility of doing something is obviously constrained by the data one can have access to, I think that this should not be used as a motivation --but rather as the best possible strategy given the circumstances.

2) The results with the applied normalisation are, according to the author, *poor*, since internationally acclaimed artists do not appear on the top. Again, I agree that an artist that did one single famous Christmas song should not be considered the most influential one in contemporary popular music. However, I also find a bit pointless the need to have the most famous artists we all know at the top of the ranking, where we expect them to be. If this is the case, why doing the entire analysis in the first place? I am obviously exaggerating here, but I believe that the answer lies somehow in between.

As such, I think that a little discussion on this in the conclusion is needed, and the normalised table could be put in the SI accompanied by a short text on its caveats.

7. PLOS authors have the option to publish the peer review history of their article (what does this mean?). If published, this will include your full peer review and any attached files.

Reviewer #1: No

Reviewer #2: **Yes: **Iacopo Iacopini

---

## [Author Response · Author response to Decision Letter 2]

26 Mar 2021

6. Review Comments to the Author

Reviewer #1: The author has addressed most of my concerns. I appreciate the work that was done to respond to both of the reviewers.

Reviewer #2: I would like to that the author again for implementing the suggested modifications, improving in this way the overall presentation of the manuscript. Based on the latest changes, I would recommend publication in PLoS One --even though I am still a bit puzzled by the entire normalisation argument.

In particular, there are two crucial points from the latest answer:

1) When the author writes "the information about the number of songs that Allmusic provides is not entirely correct. Allmusic computes the number of different recordings, even these different sessions are about the same song (...) I think that this variable is unstable and it is not suitable to normalize the covers." it seems to me that the non-suitability of the possessed data is used to justify the appropriateness of the employed methodology. While the possibility of doing something is obviously constrained by the data one can have access to, I think that this should not be used as a motivation --but rather as the best possible strategy given the circumstances.

2) The results with the applied normalisation are, according to the author, *poor*, since internationally acclaimed artists do not appear on the top. Again, I agree that an artist that did one single famous Christmas song should not be considered the most influential one in contemporary popular music. However, I also find a bit pointless the need to have the most famous artists we all know at the top of the ranking, where we expect them to be. If this is the case, why doing the entire analysis in the first place? I am obviously exaggerating here, but I believe that the answer lies somehow in between.

As such, I think that a little discussion on this in the conclusion is needed, and the normalised table could be put in the SI accompanied by a short text on its caveats.

I have added the Limitation section to discuss the normalization problem in skewed distributions and the imprecise definition of songs by Allmusic. In addition, I have added as Supplementary material a table with the top 10 ranking of artists according to the normalized impact by song and a discussion about this result.

---

## [Decision Letter · Decision Letter 3]

5 Apr 2021

Cover versions as an impact indicator in popular music: a quantitative network analysis

PONE-D-20-26285R3

Dear Dr. Ortega,

We’re pleased to inform you that your manuscript has been judged scientifically suitable for publication and will be formally accepted for publication once it meets all outstanding technical requirements.

Kind regards,

Nicola Perra

Academic Editor

PLOS ONE

Additional Editor Comments (optional):

Reviewers' comments:

Reviewer's Responses to Questions

**Comments to the Author**

1. If the authors have adequately addressed your comments raised in a previous round of review and you feel that this manuscript is now acceptable for publication, you may indicate that here to bypass the “Comments to the Author” section, enter your conflict of interest statement in the “Confidential to Editor” section, and submit your "Accept" recommendation.

Reviewer #2: All comments have been addressed

2. Is the manuscript technically sound, and do the data support the conclusions?

Reviewer #2: Yes

3. Has the statistical analysis been performed appropriately and rigorously? 

Reviewer #2: Yes

4. Have the authors made all data underlying the findings in their manuscript fully available?

Reviewer #2: Yes

5. Is the manuscript presented in an intelligible fashion and written in standard English?

Reviewer #2: Yes

6. Review Comments to the Author

Reviewer #2: All concerns and suggestions have been addressed by the author. I am happy to recommend the revised version of the manuscript for publication in PLoS One.

7. PLOS authors have the option to publish the peer review history of their article (what does this mean?). If published, this will include your full peer review and any attached files.

Reviewer #2: **Yes: **Iacopo Iacopini

---

## [Editor Report · Acceptance letter]

7 Apr 2021

PONE-D-20-26285R3 

Cover versions as an impact indicator in popular music: a quantitative network analysis 

Dear Dr. Ortega:

I'm pleased to inform you that your manuscript has been deemed suitable for publication in PLOS ONE. Congratulations! Your manuscript is now with our production department. 

Kind regards, 

on behalf of

Dr. Nicola Perra 

Academic Editor

PLOS ONE